



# Future Changes of Compound Explosive Cyclones and Atmospheric Rivers in the North Atlantic

Ferran Lopez-Marti[1,2,*], Mireia Ginesta[3,*], Davide Faranda[3], Anna Rutgersson[1,2], Pascal Yiou[3], Lichuan Wu[1], and Gabriele Messori[1,4,5]

[1]Department of Earth Sciences, Uppsala University, Uppsala, Sweden
[2]Centre of Natural Hazards and Disaster Science (CNDS), Uppsala, Sweden
[3]Laboratoire des Sciences du Climat et de l'Environnement, UMR8212 CEA-CNRS-UVSQ, IPSL & U Paris-Saclay, Gif-sur-Yvette, France
[4]Department of Meteorology and Bolin Centre for Climate Research, Stockholm University, Stockholm, Sweden
[5]Swedish Centre for Impacts of Climate Extremes (climes), Uppsala University, Uppsala, Sweden
[*]These authors contributed equally to this work.

**Correspondence:** Ferran Lopez-Marti (ferran.lopez-marti@geo.uu.se)

**Abstract.** The explosive development of extratropical cyclones and atmospheric rivers play a crucial role in driving extreme weather in the mid-latitudes, such as compound windstorm-flood events. Although both explosive cyclones and atmospheric rivers are well-understood and their relationship has been studied previously, there is still a gap in our understanding of how a warmer climate may affect their concurrence. Here, we focus on evaluating the current climatology and assessing changes

5 in the future concurrence between atmospheric rivers and explosive cyclones in the North Atlantic. To accomplish this, we independently detect and track atmospheric rivers and extratropical cyclones and study their concurrence in both ERA5 reanalysis and CMIP6 historical and future climate simulations. In agreement with the literature, atmospheric rivers are more often detected in the vicinity of explosive cyclones than non-explosive cyclones in all datasets, and the atmospheric river intensity increases in all the future scenarios analysed. Moreover, we find that explosive cyclones with atmospheric rivers are longer-

10 lasting and deeper than other explosive cyclone. Notably, we identify a significant and systematic future increase in the cylones – atmospheric river concurrences. Finally, under the worst-case scenario, the explosive cyclone – atmospheric river concurrences show an increase and model agreement over western Europe. As such, our work provides a novel statistical relation between explosive cyclones and atmospheric rivers in CMIP6 climate projections and a characterization of their joint changes in intensity and location.

## 1 Introduction

Atmospheric Rivers (ARs) are narrow and elongated corridors of horizontal moisture transport usually associated with the cold front of an extratropical cyclone (Bao et al., 2006; Ralph et al., 2004, 2017). They play an essential role in the atmospheric water vapour cycle in the mid-latitudes, accounting for 90% of the poleward moisture transport (Zhu and Newell, 1998; Guan and Waliser, 2015). Moreover, ARs drive wet and windy extreme weather events, particularly in western continental coasts,

20 such as Western Europe (Zhu and Newell, 1998; Guan and Waliser, 2015; Lavers and Villarini, 2015, 2013; Gimeno et al.,



2016). In some coastal areas, intense ARs are associated up to 95% of the time with extreme precipitation and up to 75% with wind extremes (Waliser and Guan, 2017). As a result, ARs have widespread socioeconomic impacts; for example, the majority of European storms causing insurance losses of a billion US dollars or more are linked to ARs (De Vries, 2020).

Another meteorological feature that can lead to wet and windy extremes are explosive cyclones (ECs), also known as weather bombs (Roebber, 1984; Reale et al., 2019). ECs are rapidly intensifying mid-latitude cyclones. Historically, ECs have been identified as those with a deepening rate of more than 24 hPa in 24 hours, and scaled by the latitude (Sanders and Gyakum, 1980), although this definition has been repeatedly challenged, especially for the Southern Hemisphere (Allen et al., 2010). Events such as the Presidents' Day Snowstorm of 1979 (Schultz, 2022), characterized by poor forecast accuracy, sparked extensive research into this phenomenon. Upon landfall during their intensification phase, ECs can produce widespread damage and impacts associated with strong winds, heavy precipitation, and storm surges (Fink et al., 2012; Liberato et al., 2013; Ludwig et al., 2015; Seiler and Zwiers, 2016b; Reale et al., 2019; Ginesta et al., 2023).

ECs and the associated ARs thus play a crucial role in driving extreme weather events in the mid-latitudes (Liberato et al., 2013; Davolio et al., 2020). An illustrative example is Storm Alex, an EC associated with an AR that at first produced extreme winds in France and the UK and then led to record-breaking precipitation in Italy in October 2020 (Davolio et al., 2023; Ginesta et al., 2023). The climatological relationship between ECs and ARs has been previously studied and the literature evidences that ARs are more often found in the surroundings of EC than non-ECs (Eiras-Barca et al., 2018; Zhang et al., 2019; Guo et al., 2020). The physical reason behind this is that ARs incorporate moisture into cyclonic systems, leading to a faster and stronger deepening of the cyclone (Zhu and Newell, 1994; Ferreira et al., 2016). In this context, an important process for explosive cyclogenesis is the latent heat release from the moisture brought by the AR. Cyclones with ARs show larger moisture inflow but do not show significant differences in low-level baroclinicity nor upper-level potential vorticity, suggesting that diabatic process are the main contributors to explosive intensification (Pinto et al., 2009; Zhang and Ralph, 2021).

A range of studies have investigated the impact of anthropogenic climate change on Extratropical Cyclones and ARs individually (Lavers et al., 2015; Zappa et al., 2013). The thermodynamic response of ARs to climate change is characterized by an increase in Integrated Water Vapor Transport (IVT). This increase is driven by the Clausius-Clapeyron relation, which implies a rise in moisture content in a warmer atmosphere. However, the vertically integrated water vapor content undergoes further amplification compared to surface water vapor (Payne et al., 2020). This thermodynamic signal would act to increase the number of ARs detected in a warmer climate (Thandlam et al., 2022; Espinoza et al., 2018; Zhang et al., 2024; O'Brien et al., 2022; Wang et al., 2023). Similarly, the thermodynamic response acts to increase the precipitation within extratropical cyclones (Yettella and Kay, 2017). The dynamical response to climate change, such as changes in atmospheric circulation patterns, is more uncertain (Shepherd, 2014). In the North Atlantic, dynamic changes are mostly driven by changes in the eddy-driven jet, which serves as a guide for extratropical cyclones. The tug of war between the upper tropospheric warming and the Arctic amplification leads to a high uncertainty regarding the changes in the jet over the North Atlantic and western Europe (Shaw et al., 2016). However, climate models indicate a decline in the number of extreme extratropical cyclones in the North Atlantic, and a local increase over the North Sea in EC frequency (Priestley and Catto, 2022; Zappa et al., 2013; Seiler



and Zwiers, 2016a). Regarding ARs, studies also point to an increase in frequency, intensity, and size in western Europe and a
northward shift of the AR location and landfall (Lavers et al., 2013; Ramos et al., 2016; Gao et al., 2016; Zhang et al., 2024).

The hazards associated with the joint occurrence of explosive cyclones and atmospheric rivers, especially along the western
coast of Europe, underscores the importance of evaluating the projected changes in their concurrence in future climates. In this
study, we assess future projections of the interplay between ECs and ARs using state-of-the-art CMIP6 data. We specifically

evaluate the frequency of EC and AR concurrence in the present climate for the ERA5 reanalysis and CMIP6 models, and in
three end-of-century scenarios for the CMIP6 models. Moreover, we also assess the future changes in the intensity and location
of such compound events.

This manuscript is structured as follows: Section 2 describes the datasets employed in this study, while Section 3 explains the
methodologies for tracking ARs and cyclones, and the calculation of their concurrences. In Section 4 we evaluate and discuss

the performance of the CMIP6 models with ERA5 reanalysis. Section 5 shows the results and discussion of the future changes
in frequency, intensity, and location of this compound event. Finally, Section 6 summarizes the key findings and provides the
conclusions of this study.

## 2   Data

We use the ECMWF reanalysis ERA5 (Hersbach et al., 2020) with a horizontal resolution of 0.25º × 0.25º as an observationally-

constrained reference for the current climate and validation of the Global Climate Models (GCMs). For the AR detection, the
variables used are specific humidity, meridional and zonal wind components at 1000, 925, 850, 700, 600, 500, 400 and 300
hPa (Section 3.2). For cyclone detection, the variable used is the sea level pressure (SLP). In both cases, the variables are at 6
hourly resolution during the extended winter period (October to March) between 1980 and 2009 for the North Atlantic region
[25-65ºN; 80°W-10°E].

Further, we use one member each of six different GCMs from the CMIP6 dataset (Eyring et al., 2016): MPI-ESM1-2-
LR, MPI-ESM1-2-HR, NorESM2-MM, EC-Earth3, CMCC-ESM2, MIROC6. More detailed information about the GMCs
used can be found in the Appendix A1. We evaluate the listed GCMs for two periods: the current climate (1980-2009) using
historical simulations and the future climate at the end of the 21st century (2070-2099), where we used simulations following
the SSP1-2.6; SSP2-4.5 and SSP5-8.5 forcing scenarios (Riahi et al., 2017). Variables used for tracking ARs and cyclones in

the GCMs are the same as used for ERA5. The current list of GCMs used in this study is limited by the availability of 6-hourly
instantaneous variables for the historical and the three scenarios experiments in hybrid-sigma pressure model levels, which are
required to interpolate to the necessary pressure levels for the IVT calculation when detecting ARs.



## 3   Methods

### 3.1   Extratropical cyclone tracking

There are several detection and tracking algorithms for extratropical cyclones, most of them using either SLP or lower-tropospheric vorticity (Neu et al., 2013). Here, we detect and track cyclones based on SLP using the TempestExtreme algorithm developed by Ullrich et al. (2021). This command-line software facilitates adaptable and rapid feature detection and tracking for extratropical cyclones but also for ARs.

To identify extratropical cyclones, we recognize candidate "nodes" corresponding to local minima in the SLP field. Nodes
within a $6°$ great circle distance (GCD) of each other are merged. Next, to connect these candidate nodes into tracks the distance between consecutive detections should not exceed 6 GCD degrees. The tracks must persist for a minimum of 24 hours, the maximum duration between two detections is set at 6 hours, and the cyclones must have moved at least 12 GCD degrees to filter out stationary lows, such as the Icelandic Low. The relevant codes are available in the Appendix A2.

In addition, extratropical cyclones are classified as ECs if their Normalized Deepening Rate (NDR), as defined by Sanders
and Gyakum (1980), is equal to or higher than 1:

$$NDR_c = \frac{DR_{24h}}{24h}\frac{\sin(60°)}{\sin(\varphi)} \tag{1}$$

where $DR_{24h}$ is the pressure difference over 24 hours measured at the storm centre and $\varphi$ is the latitude at its second-time step. Cyclones that do not fulfil this condition are classified as non-ECs. Table A2 in Appendix A4 lists the number of ECs and non-ECs tracked in each model and scenario and in ERA5. The number of cyclones detected fits inside the range number
of cyclones detected in Neu et al. (2013) using several cyclone tracking algorithms. Figure 1 shows the storm track density for ECs and non-ECs in ERA5 and CMIP6 models. These results agree with Priestley and Catto (2022) and Zappa et al. (2013) track density despite using different tracking algorithms. Our tracking method shows some differences between ERA5 and CMIP6 models (Figure 1c,d). In the models, there are fewer ECs and non-ECs around the south tip of Greenland, Iceland, west Scandinavia and the Gulf of Genoa in the Mediterranean Sea. In particular, for ECs the CMIP6 models underestimate track
density in the northern part of the domain, whereas for the non-EC CMIP6 show a northward shift of the storm track with lower track density in the south and higher track density in the north of the domain.





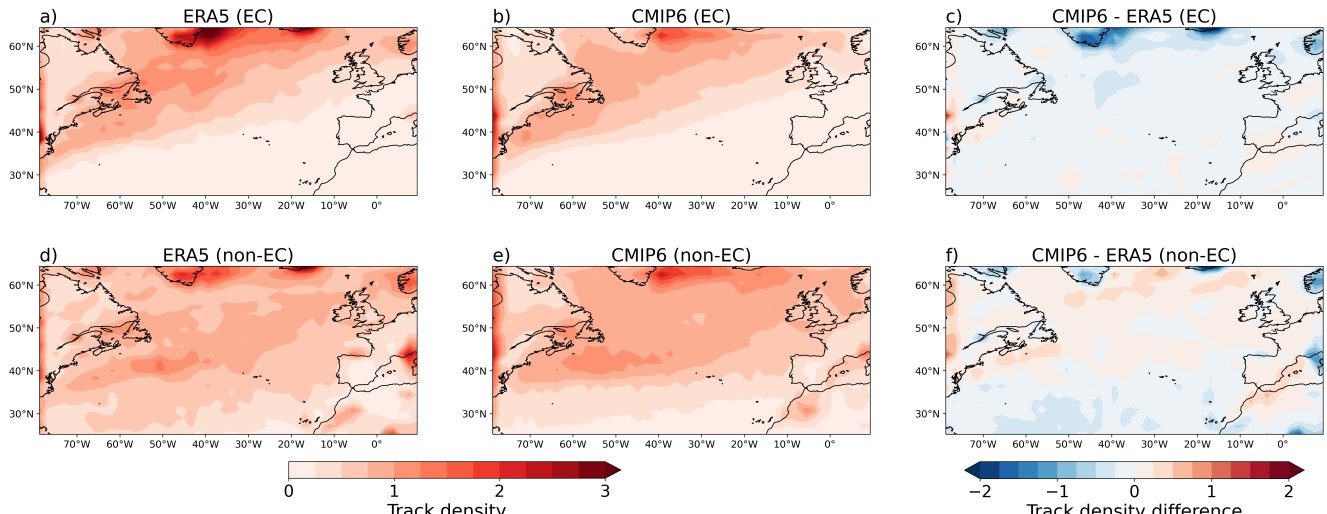

**Figure 1.** EC track density climatology [1980-2009] for ERA5 (a), historical simulations of CIMP6 models (b), and difference between ERA5 and CMIP6 (c). In panels (d,e,f) same as (a,b,c) but for non-ECs. Units are the number of cyclones per $1.5°$ spherical cap per month.

## 3.2 AR detection and tracking

There are two primary approaches to detecting ARs: one involves using the Integrated Water Vapor, commonly applied to satellite data, while the other, more broadly used, consists in computing the IVT (Gimeno et al., 2014; Shields et al., 2018). In this study, we calculate the IVT for both reanalysis and model data. The IVT is defined for each grid point as:

$$IVT = \left[ \left( \frac{1}{g} \int_{1000hPa}^{300hPa} qu\,dp \right)^2 + \left( \frac{1}{g} \int_{1000hPa}^{300hPa} qv\,dp \right)^2 \right]^{1/2} \tag{2}$$

where $q$ is the specific humidity, $u$ is the zonal wind component, $v$ is the meridional wind component, and $g$ is the gravitational acceleration. Moreover, we separately compute the eastward ($IVT_E$) and northward ($IVT_N$) components of IVT, which correspond to the two terms inside of the brackets in Eq. (2) respectively.

For the detection of ARs, we also use the TempestExtreme algorithm developed by Ullrich et al. (2021). We find candidates for atmospheric rivers by detecting ridges in the IVT field. Ridges are defined as points where the Laplacian of the IVT is below $-4\times10^4$ kg m$^{-2}$ s$^{-1}$ rad$^{-2}$, as this operator identifies elongated areas and regions of local minima. In addition, the IVT should be higher than $250$ kg m$^{-1}$ s$^{-1}$. Each candidate should have an area larger than $4 \times 10^5$ km$^{-2}$. The detected candidates are concatenated if at least one grid point is detected as AR in sequential timesteps. In addition, they should last 60 hours. This AR tracking methodology is less sensitive to a generalized increase of IVT in the future climate due to the Clausius-Capeyron relationship because we detect AR candidates using the Laplacian of the IVT instead of the IVT field. Thus, AR candidates are detected when having a pronounced gradient of IVT (not dependent on the background IVT), as discussed further in Section 5,





this may affect the detected changes in number of AR tracks in future climates. The relevant codes are available in Appendix
A3 and the detailed number ARs tracks detected are reported in Table A3 in Appendix A4. Figure 2a shows the frequency
of ARs as the percentage of time-steps detected with an AR for ERA5. Our tracking methodology reproduces well the AR
climatology when compared with Guan and Waliser (2015). The AR climatological frequency for the historical simulations
of the CMIP6 models is larger when compared to ERA5 and shows a shift south. In other words, more ARs are detected in
CMIP6 and those tracks are chiefly located in the lower midlatitudes (Figure 2b,c).

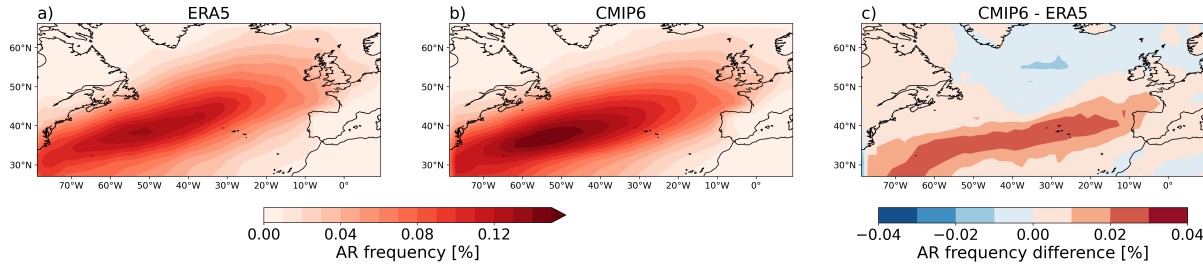

**Figure 2.** AR frequency climatology [1980-2009] for ERA5 (a), historical simulations of CIMP6 models (b), and difference between ERA5
and CMIP6 (c). Units are the percentage of time-steps detected as AR.

## 3.3 Concurrences

For each extratropical cyclone (both ECs and non-ECs) we compute the Maximum Deepening Point (MDP), which is the
maximum difference in SLP between two consecutive 6-hourly timesteps. This metric allows us to evaluate the influence of
ARs on the development of the cyclone before and after its maximum intensification.

Subsequently, we determine whether a specific timestep of an extratropical cyclone (EC or non-EC) is linked to an AR by
detecting the presence of an AR within a 1500 km of the cyclone center.

Figure 3 shows an example of our detection methodology applied to storm Xynthia. EC Xynthia underwent rapid intensifica-
tion before making landfall on February 27, 2010. It caused widespread damage across Western European countries, specially
France and Spain. In Figure 3, the black dots represent the cyclone's path, while the red crosses indicate the time steps concur-
rent with the presence of an AR. The shaded areas depict the regions identified as ARs at each concurrence time step during
the cyclone trajectory. Xynthia was associated with an AR during its intensification phase until landfall, suggesting that the AR
may have contributed to its intensification (Liberato et al., 2013).





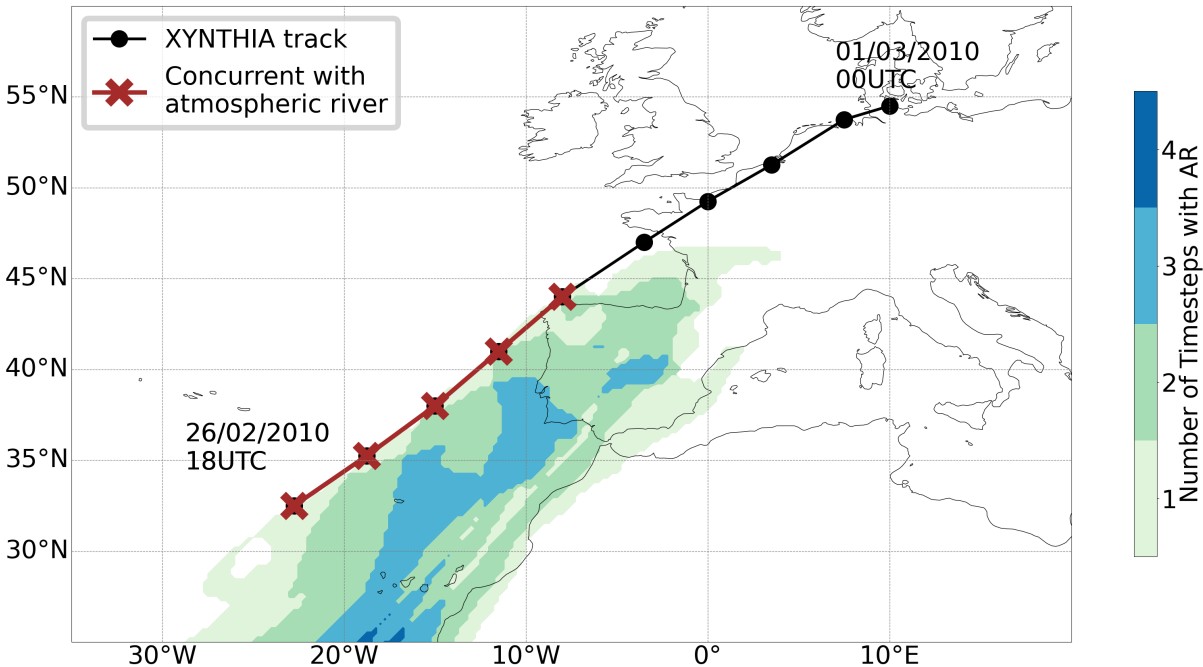

**Figure 3.** Example of detection and tracking of EC concurrent to an AR in ERA5. Red crosses mark the location of the cyclone at times when it concurred with an AR. Black dots indicate no concurrence. Shaded areas depict the regions identified as ARs at each concurrence time step during the cyclone trajectory

## 4 Concurrences in Present Climate

In this section, we evaluate the concurrence of ECs and ARs in the ERA5 reanalyses and we compare it with those obtained in climate models. Figure 4a shows the rate of coincidence between ECs and ARs, indicating the fraction of ECs that are associated with an AR (see Section 3.3), as a function of time from the MDP point. ERA5 shows a maximum rate of coincidence at around 6 hours after the MDP point. This maximum is about 0.65, meaning that 65% of the ECs are associated with ARs. The rate of coincidence is minimum at the initial stages of cyclone formation, with 45% of the ECs associated with an AR. During the dissipation stages of the cyclones, around 55% of ECs are associated with an AR. In other words, the rate of coincidence is higher after the MDP, when the ECs are deeper (see Section 5.3). This suggests that a mature cyclone, when its SLP is lower, favours the detection of an AR in its surroundings. These results agree with the ones obtained by Eiras-Barca et al. (2018) despite a shift of 6-12 hours in the concurrence peak, which comes from how the MDP is calculated (See Section 6). The curves of the CMIP6 models show a similar shape to that of ERA5, with a maximum rate of coincidence centred 6/12 hours after the MDP, a minimum at the formation of the cyclone, and a secondary minimum at the dissipation stage. Some of the models, specifically CMCC, MPI-HR, and MPI-LR, overestimate the rate of coincidence for almost the whole lifetime of the ECs. CMCC and MPI-LR show a maximum of 75% of ECs associated with an AR, which is 10% more than ERA5. On the





contrary, EC-EARTH3, MIROC6, and NorESM underestimate the rate of coincidence before the MDP, while they overestimate it, especially MIROC6, after the MDP. EC-EARTH3 and NorESM are the models that are more similar to the reanalysis, with maximum biases of around 0.05 along the lifetime of the ECs. Figure 4c shows the inter-seasonal variability as the standard deviation of the rate of coincidence between ECs and ARs over the 30 winter seasons. CMIP6 models and ERA5 show an inter-seasonal variability between 0.07 and 0.1, being a bit higher on the first and last time steps from the MDP. CMIP6 models

reproduce similar concurrence variability as ERA5 and differences in concurrence rate between them and ERA5 are within the internal variability.

Figure 4b shows the rate of coincidence of non-ECs and ARs. ERA5 shows a much flatter curve than for of the ECs. There is a maximum in the coincidence rate at the MDP, around 50%, with two minimums at the formation and dissipation stages of the cyclones, approximately 45%. Thus, there is a relatively small variation in the coincidence rate throughout the lifetime of

the non-ECs. These results for non-ECs also agree with the ones obtained by Eiras-Barca et al. (2018). All models resemble ERA5. However, CMCC overestimates the coincidence rate throughout the lifetime of the cyclones by around 0.05. On the contrary, EC-EARTH3 underestimates this rate by approximately 0.05. The models that perform better, especially around the MDP, are MPI-LR, MIROC6, and NorESM. They nonetheless underestimate or overestimate the rate by around 0.03 at the formation and dissipation stages. Figure 4d shows the inter-seasonal variability for the rate of coincidence between non-ECs

and ARs. In this case, CMIP6 models and ERA5 show an inter-seasonal variability between 0.09 and 0.12, showing larger values on the first and last time steps for some models. CMIP6 models reproduce similar concurrence variability as ERA5 and the differences in concurrence rate between them and ERA5 are within their internal variability.

In summary, for ECs, maximum model biases occur during the MDP, reaching around 0.08. Even though this represents almost 15% of the ERA5 value, the internal variability of the datasets is slightly larger. For non-ECs, models have biases

particularly during the formation and dissipation stages of cyclones, peaking at around 0.05, or around 11% of the ERA5 value, but again smaller than the internal variability. Overall, the models reproduce the qualitative features of the life cycle of the rate of coincidence between both ECs and non-ECs and ARs. The quantitaive differences with ERA5 are generally smaller than the datasets' internal variability.







**Figure 4.** Rate of coincidence between explosive cyclones (ECs) and atmospheric rivers (ARs) (a) and non-explosive cyclones (non-EC) and ARs (b) for ERA5 and the historical runs of CMIP6 models [1980–2009]. Inter-seasonal variability as the standard deviation of the rate of coincidence over the 30 winter seasons between ECs and ARs (c) and non-ECs and ARs (d).

# 5 Future projections

In this section, we analyze changes in the concurrence of cyclones and ARs in CMIP6 models between the historical simulations (1980-2009) and the three scenario simulations (2070-2099).

## 5.1 Changes in concurrence frequency

Figure 5a,b show a clear increase in the rate of coincidence of ARs with ECs and non-EC events across all warming scenarios. In the SSP5-8.5 scenario, the multi-model mean rate of coincidence increases by approximately 12% throughout the lifetime of



both classes of cyclones. For the SSP1-2.6 and SSP2-4.5 scenarios, the increases are around 5% and 9% for ECs, respectively, and 4% and 8% for non-EC events.

Table A2 in Appendix A4 shows the number of ECs and non-ECs tracks and table A3 shows the number of AR tracks detected for ERA5 and the CMIP6 models in each scenario. Hence, these tables also show the changes in the occurrence of the features individually in future scenarios compared to historical ones. The number of ECs decreases in future scenarios with

respect to the historical for all models, usually with the largest decrease for the worst-case scenario (SSP5-8.5). There is little change in the number of non-ECs, although most of the models also project a small decrease in future scenarios. Regarding the number of ARs, for almost all the models there is a small increase in the number in the worst-case scenario. However, the changes are not linear, that is, some of the models (MPI-LR, MPI-HR, NorESM, MIROC6) depict a larger number of ARs in the SSP1-2.6 scenario than in the SSP5-8.5. The lower-than-expected increase in the number of AR tracks in the warmer

scenarios at the end of the century could be explained by the nature of the AR tracking, which detects ARs in the Laplacian of the IVT (Section 3.2). However, Table A3 only shows the number of AR tracks and does not account for the AR duration or extension, which might result in an increase of AR activity under a warmer climate (Zhang et al., 2024; O'Brien et al., 2022). Despite not finding a strong increase of ARs' tracks, the total number of times an AR is detected in the surroundings of both types of cyclones nonetheless shows a clear increase in all scenarios, with the increase becoming larger with higher warming.

In addition, the absolute number of compound events of ECs or non-ECs together with ARs also increases in all scenarios (not shown). Hence, the combination of a small increase in AR frequency and a small decrease in the number of ECs and non-ECs results in an increase in the ratio of coincidence with the level of warming (Figure 5a,b). This points to changes in the characteristics of the ARs or cyclones, or of their interactions, as drivers of the identified changes, rather than these being a simple statistical effect of more individual cyclones and ARs.

The spread across CMIP6 models is the primary source of uncertainty when evaluating changes in the rate of coincidence, and exhibits overlap between different scenarios (Fig. 5a, b). The choice of model thus has a significant impact on the results when quantifying the increase in cyclone-AR compound events. Nevertheless, the SSP5-8.5 scenario's inter-model spread is well-separated from that of the historical period for both ECs and non-ECs (Fig. 5a, b). The inter-model spread further exceeds inter-seasonal variability for the models we analyse (cf. Fig. 5a, b and 5c, d). Indeed, the inter-seasonal variability of the

model ensemble is consistently smaller than the multi-model mean differences between any of the three warming scenarios and historical for ECs (Fig. 5c), and systematically smaller than the multi-model mean differences for non-ECs in the SSP5-8.5 scenario and, at most lead times, for the SSP2-4.5 scenario (Fig. 5d). Our results thus point to anthropogenic climate change exerting a strong influence on the increase in coincidence rates of cyclones and ARs, which is most evident for higher warming levels and, for the latter, cannot be attributed to natural variability or inter-model spread.







**Figure 5.** Rate of coincidence for the historical runs and ERA5 [1980–2009] and the future scenarios runs [2070–2099] between ECs and ARs (a) and non-EC and ARs (b). Solid lines show the multi-model mean, shape-points individual models and shades the inter-model spread. The difference in the multi-model mean rate of coincidence between the forcing scenarios and historical runs for ECs and ARs (c) and non-EC and ARs (d). Solid lines show the multi-model mean difference, shades show the inter-seasonal variability of the multi-model ensembles in future scenarios and historical, and the dashed line the inter-seasonal variability for ERA5.

## 5.2 Changes in AR intensity

The intensity of ARs can be quantified in different ways, here we use the maximum value of IVT (IVT-max) within the detected AR. We use this metric because it represents a good proxy for the total transport of moisture of an AR and is not dependent on the boundaries of the detected AR (Ralph et al., 2017). Moreover, the IVT-max is usually located at the core of the AR, this makes this variable also a good proxy for AR extension and duration as ARs reaching higher IVT values at their core tend to be larger and last longer (Guan et al., 2023). Our results indicate that the IVT-max of ARs associated with ECs and non-ECs



increases proportionally to the warming in future scenarios (Figure 6). When comparing the IVT-max between scenarios, it is particularly relevant that under the SSP5-8.5 the multi-model mean remains above 1250 kg·m·s$^{-1}$ for more than 48h, this means that on average ARs under such conditions will be primarily hazardous according to the AR-scale by Eiras-Barca et al. (2021). For ARs with non-EC events, the IVT-max is constant throughout the entire cyclone lifetime, suggesting a uniform

inflow of atmospheric moisture transport. On the other hand, for ARs associated with ECs, the IVT-max reaches its peak around the MDP of the cyclone's lifecycle, showing the maximum moisture transport during the cyclone's most active phase (Figure 6a,b). The AR intensity for ERA5 is larger than any model for the historical period because ERA5's resolution is almost 4 times higher than the CIMP6 models, and attains larger values of IVT-max. Despite this, the ERA5 and CMIP6 curves show the same qualitative behaviour during the cyclones' lifetime and the inter-seasonal variability in both datasets is of the same

order of magnitude (Figure 6).

When evaluating uncertainties for changes in the AR intensity, inter-seasonal variability is slightly smaller than the inter-model spread. Both are dependent on model resolution, as the IVT-max magnitude is sensitive to resolution and all datasets are treated on their native grids. For both ARs with ECs and non-ECs, the inter-model spread of both the SSP2-4.5 and the SSP5-8.5 scenarios is well-separated from that of the historical period (Fig. 6a, b). Moreover, the difference in the multi-model mean

between the future scenarios and the historical is more pronounced than the inter-seasonal variability for all three scenarios and for both ECs and non-ECs (Figure 6c,d). Thus, the IVT-max increases in all scenarios and our results point to this being larger than both inter-model spread and internal model variability in most scenarios, and thus ascribable to anthropogenic warming.





**Figure 6.** Mean IVT-max for the historical runs and ERA5 [1980–2009] and the future scenarios runs [2070–2099] of ARs associated with ECs (a) and ARs associated with non-EC (b). Solid lines show the multi-model mean, shape-points individual models and shades the inter-model spread. The difference in the multi-model mean IVT-max between the forcing scenarios and historical run for ECs and ARs (c) and non-EC and ARs (d). Solid lines show the multi-model mean difference, shades show the inter-seasonal variability of the multi-model ensembles in future scenarios and historical, and the dashed line the inter-seasonal variability for ERA5.

## 5.3 Changes in explosive cyclone intensity

In this section, we examine the cyclone core SLP to assess changes in the intensity of ECs and non-ECs with the presence or not of ARs under climate change. The presence of ARs influences the evolution of SLP through the life-cycle of the two types of cyclones differently, but in both cases when an AR is present the cyclone is deeper (Fig. 7e, f). The concurrence of ECs with ARs makes the ECs substantially deeper, especially before the MDP, where ECs are between 7.5 hPa and 15 hPa deeper. After the MDP the influence of the AR on the cyclone intensity decreases. Thus, ARs play an important role before





the EC MDP, making these storms deeper from an earlier stage and resulting in longer-lasting and more intense cyclones. The
particular influence in the first half of the cyclone life span suggests that moisture brought by the AR plays a key role in the
EC intensification. On the other hand, the non-ECs also get deeper when occurring together with an AR, with a deepening
between 2 and 7 hPa, having its peak also just before the MDP. The results from ERA5 show the same behaviour for both types
of cyclones but with lower intensity (Fig. 7b,d).

The climate change influence on cyclone intensity for any of the four types of compound event is very limited. All three
forcing scenarios show the same life-cycle behaviour and intensity changes as the historical ensemble. Figure A1 shows that the
difference of the multi-model mean between the forcing scenarios and the historical is negligible (less than 1 hPa). It is slightly
larger only for the ECs without ARs, probably due to the reduced number of these events (but as a result their inter-seasonal
variability is also larger). The <1 hPa shift can be compared to the models' inter-seasonal variability of between 1 and 2 hPa
(Fig. A1). Similarly, the multi-model spreads of all scenarios are about ± 4 hPa (Fig. 7). Thus, the models do not show a robust
change in the cyclone intensity under climate change scenarios.





**Figure 7.** Mean SLP for the historical runs and ERA5 [1980–2009] and the future scenarios runs [2070–2099] of ECs with ARs (a), non-EC with ARs (b), ECs without ARs (c) and non-ECs without ARs (c). Solid lines show the multi-model mean, shape-points individual models and shades the inter-model spread. The difference in the multi-model mean SLP between the ECs with AR and without AR (e) and non-EC with AR and without AR (f).



## 5.4 Changes in location of concurrent ECs and ARs

To assess the spatial distribution of changes observed across the North Atlantic, we show the rate of EC-AR coincidence over a 3-degree spherical cap. In the historical period, the rate is highest along the climatological North Atlantic storm track, extending from the western to the northeastern part of the basin (Fig. 8a,b). CMIP6 models reproduce closely this spatial pattern. Differences between future projections and the historical period show an increase in the rate of coincidence, all the stronger for higher warming levels. The worst-case scenario (SSP5-8.5) shows the largest increase and a high model agreement across almost the entire domain. The southern part of the domain shows a noisier pattern and weaker model agreement due to the reduced number of events in the area. Only the worst-case scenario shows a clear increase in the rate of coincidence over Europe, showing model agreement over the British Isles, northern France and the Iberian peninsula. In other words, a larger proportion of landfalling ECs together with ARs at their maximum deepening point is expected under the SSP5-8.5 scenario. This highlights the possibility of an increase in wet and windy extremes in Western Europe in a high-emissions future.

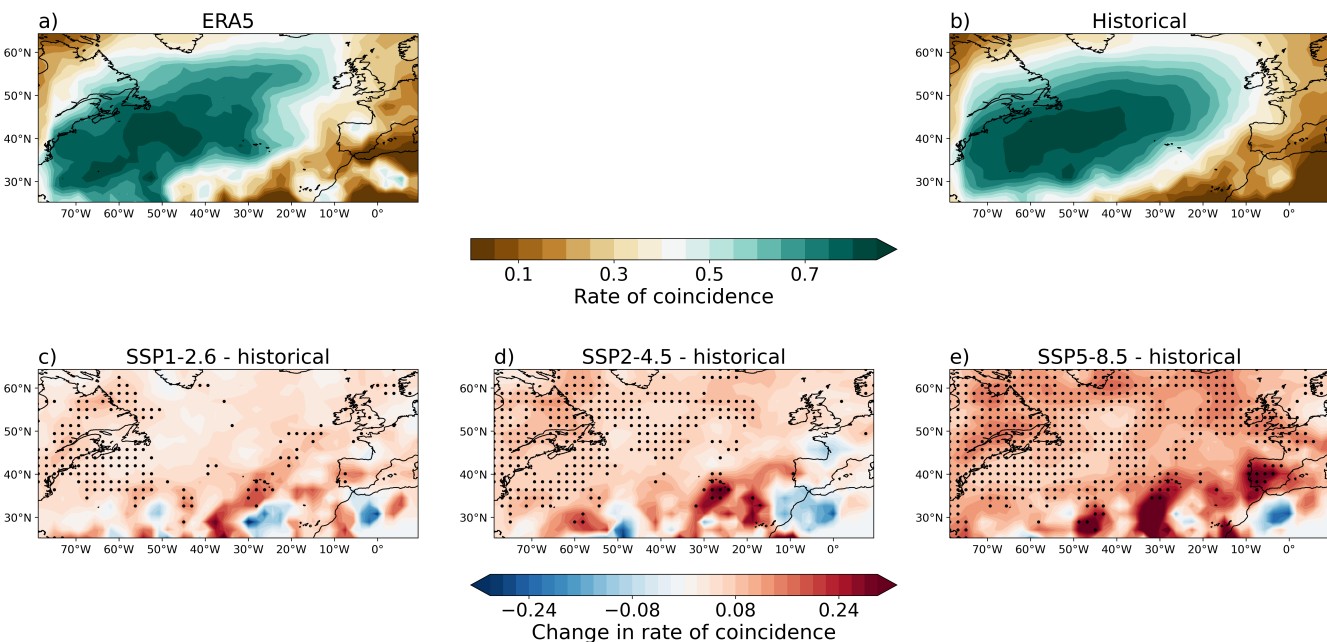

**Figure 8.** Rate of coincidence between ECs and ARs for ERA5 (a), and the multimodel-mean of the CMIP6 historical runs (b). Differences between future projections and historical periods of the three scenarios: SSP1-2.6 (b), SSP2-4.5 (c), and SSP5-8.5 (d). Dots denote where all the CMIP6 models agree on the sign of change.





## 6 Conclusions

We have used six Global Climate Models (GCMs) participating in CMIP6 to evaluate the change in the concurrences of cyclones and Atmospheric Rivers (ARs) in three different future scenarios under climate change in the North Atlantic. We

have evaluated the performance of the models using the ERA5 reanalysis. Our main findings are summarized as follows:

- For the present period, in ERA5 nearly 65% of the Explosive Cyclones (ECs) are associated with an AR within the 6 hours following the Maximum Deepeing Point (MDP). The fact that AR concurrences are larger after the MDP suggests that mature ECs (when they are deeper) can facilitate the formation of ARs in their surroundings. Despite some biases in the magnitude of the coincidence rate, CMIP6 models exhibit qualitatively similar concurrence rates to ERA5, peaking 6

to 12 hours after the MDP. Conversely, the evolution of the concurrence rate over the cyclone's lifespan is less pronounced for non-Explosive Cyclones (non-ECs).

- In future scenarios, there is an increase in the rate of coincidence between ECs and ARs, with the magnitude of the increase proportional to the level of warming. All models agree on the sign of the change. In the worst-case scenario (SSP5-8.5), there is a maximum increase in the rate of coincidence from 0.68 to 0.8. Even in the best-case scenario

(SSP1-2.6), there is an increase of around 6%. For non-ECs, there is an increase in the rate of coincidence, ranging from 5% to 12% across scenarios. The increase in SSP5-8.5 and SSP2-4.5 scenarios is larger than the inter-seasonal model variability for both ECs and non-ECs, but the inter-model spread of the historical period is only well-separated from that of the SSP5-RCP8.5 scenario.

- In all warming scenarios there is an increase in AR intensity. This is larger than the inter-seasonal variability for ARs

associated with both ECs and non-ECs. Moreover, the inter-model spread of both the SSP2-4.5 and the SSP5-8.5 scenarios is well-separated from that of the historical period. Under the SSP5-8.5 scenario, the maximum Integrated Vapour Transport (IVT) of ARs associated with ECs is projected to exceed on average $1250\,\mathrm{kg\,m^{-1}s^{-1}}$ for more than 48 hours, indicating exceptional and hazardous AR conditions.

- ECs are deeper when they are associated with an AR, especially before the MDP. This indicates that ECs with an AR

get deeper earlier compared to ECs without ARs, making these cyclones deeper for a longer time and potentially more hazardous. Non-ECs are also deeper when having an AR in their surroundings but their intensification is lesser compared to the ECs with ARs.

- The concurrence of ARs with ECs will be more frequent in the North Atlantic basin in future climates. There is an increase in both agreement among models and magnitude of the change with the degree of warming, with the SSP5-8.5

scenario showing the largest change. Under this most severe scenario, Europe is exposed to this increase, particularly the southern Iberian peninsula, the British Isles, France and Scandinavia.

Our results for concurrence rates of cyclones and ARs in the present climate are broadly consistent with Eiras-Barca et al. (2018), despite the latter study using the lower resolution reanalysis ERA-Interim (Dee et al., 2011), different tracking and



selection algorithms and a slightly different historical analysis period. The main difference in our results is in the peak of concurrence for ECs, which in Eiras-Barca et al. (2018) is at the MDP. The latter authors calculated the MDP using a 24-hour time difference, whereas we calculated it with a 6-hour difference. Thus, our methodology can provide a more accurate MDP, which may explain the above discrepancy. Previous studies found an increase in the IVT of ARs under climate change (Payne et al., 2020), as well as an increase in their frequency (Espinoza et al., 2018; Wang et al., 2023; Ramos et al., 2016). These results align with the increase of IVT-max detected for ARs associated with ECs and non-ECs and the increase of concurrences between cyclones and ARs, partly driven by an enhanced AR frequency. Moreover, other studies have found an increase in the frequency and severity of extratropical cyclones under climate change, mainly over the British Isles, and an eastward extension of the storm track activity over Europe (Priestley and Catto, 2022; Zappa et al., 2013; Seiler and Zwiers, 2016a). Our results show a generalized increase in compound events of ECs with ARs in most of the North Atlantic basin. A robust increase in concurrence over the British Isles, Iberia and north France is only observed under the most severe climate change scenario.

Our analysis has limitations that should be acknowledged. The main constraint is the reduced number of CMIP6 models and members used. The number of models or ensemble members used is limited by the availability of data on multiple vertical levels at the 6-hourly resolution, necessary to compute IVT. We chose all models and members from CMIP6 where these variables were available for the historical, SSP1-2.6, SSP2-4.5 and SSP5-8.5 experiments. In particular, we deemed it important to go beyond the worst-case scenario (SSP5-8.5), and also look at the implications of lower warming levels. A further caveat is that we used a single tracking algorithm, namely the TempestExtreme software (Ullrich et al., 2021). While we have compared our results for the present period with a previous study (Eiras-Barca et al., 2018), this does not detract from the fact that our results depend on the detection and tracking method (Neu et al., 2013). Furthermore, future studies should aim to delve deeper into isolating the dynamic signal from the thermodynamics of the climate change response. Finally, future work should explore future changes in the wet and windy extremes associated with the compound meteorological events investigated here.

*Code availability.* The scripts are available upon reasonable request.

*Data availability.* ERA5 data are available on the C3S Climate Data Store at https://cds.climate.copernicus.eu/#/home. CMIP6 data are available on the ESGF Metagrid web application at https://aims2.llnl.gov/search.





**Appendix A**

**A1    CMIP6 models information**

**Table A1.** Description of the CMIP6 models and member used for the historical, SSP1-2.6, SSP2-4.5 and SSP5-8.5 scenarios.

| Model Name | Member | Resolution | Reference |
|---|---|---|---|
| MPI-ESM1-2-LR | r1i1p1f1 | T63 spectral truncation (∼200 km): 192 x 96 longitude/latitude; 47 vertical levels (top level 0.01 hPa) | Mauritsen et al. (2019) |
| MPI-ESM1-2-HR | r1i1p1f1 | T127 spectral truncation (∼100 km): 384 x 192 longitude/latitude; 95 vertical levels (top level 0.01 hPa) | Müller et al. (2018) |
| NorESM2-MM | r1i1p1f1 | 1.25° × 0.9375° regular grid: 288 x 192 longitude/latitude; 32 vertical levels (top level 3.6 hPa) | Seland et al. (2020) |
| EC-Earth3 | r1i1p1f1 | T255 spectral truncation (∼80km): 512 x 256 longitude/latitude; 91 vertical levels (top level 0.01 hPa) | Döscher et al. (2022) |
| CMCC-ESM2 | r1i1p1f1 | Regular grid 0.9° × 1.25°: 288 x 192 longitude/latitude; 30 vertical levels (top level 2 hPa) | Cherchi et al. (2019) |
| MIROC6 | r1i1p1f1 | T85 spectral truncation (∼160 km): 256 x 128 longitude/latitude; 81 vertical levels (top level 0.004 hPa) | Tatebe et al. (2019) |

**A2    TempestExtremes Code for Detecting and Tracking Extratropical Cyclones**

To identify extratropical cyclones, we use the executable *DetectNodes*, which recognizes candidate "nodes" corresponding to local minima in the SLP field. Subsequently, we employ *StitchNodes* to connect these candidate nodes into tracks. The tracking codes set-up with the parameters used are as follows:

```
DetectNodes                             StitchNodes
-in_data "data_slp"                     -in_data "detect_nodes_output"
-out "detect_nodes_output"              -out "cyclone_tracks"
-searchbymin slp                        -in_fmt "lon,lat,PSL"
-mergedist 6.0                          -range 6.0
-minlon -80.0                           -mintime 24h
-maxlon 10.0                            -maxgap 6h
-minlat 25.0                            -min_endpoint_dist 12.0
-maxlat 65.0                            -out_file_format "csv"
-regional
-outputcmd "slp,min,0"
```

**A3    TempestExtremes Code for Detecting and Tracking ARs**

For the detection of ARs we use the executable *DetectBlobs* and to connect candidates or "blobs" we use the executable *StitchBlobs* with the following parameters:



```
 ./DetectBlobs                              ./StitchBlobs
-in_data "data_IVT"                        -in "detect_blobs_output"
-out "detect_blobs_output"                 -out "ar_tracks"
–latname LAT                               -latname LAT
–lonname LON                               -lonname LON
-thresholdcmd "_LAPLACIAN{8,10}            -var "binary_tag"
(_VECMAG(UQ_FLUX,VQ_FLUX)),<=,-40000,0;    -mintime 10
_VECMAG(UQ_FLUX,VQ_FLUX),>=,250,0"         -regional
-geofiltercmd 'area,>=,4e5km2'
–minlat 25
–minabslat 15
–minlon –80
–maxlon 10
–maxlat 65
-regional
```

## A4 Number of Cyclones and ARs detected in ERA5 and CMIP6

**Table A2.** Number of EC and non-EC tracks detected in each dataset.

|  | EC tracks | | | | non-EC tracks | | | |
|---|---|---|---|---|---|---|---|---|
|  | Historical | SSP1-2.6 | SSP2-4.5 | SSP5-8.5 | Historical | SSP1-2.6 | SSP2-4.5 | SSP5-8.5 |
| ERA5 | 1591 | - | - | - | 3100 | - | - | - |
| MPI-LR | 1072 | 1007 | 958 | 939 | 2273 | 2255 | 2071 | 1991 |
| MPI-HR | 1342 | 1304 | 1229 | 1186 | 2772 | 2664 | 2703 | 2670 |
| EC-Earth3 | 1471 | 1269 | 1282 | 1306 | 2600 | 2728 | 2635 | 2544 |
| NorESM2-MM | 1427 | 1323 | 1356 | 1360 | 3487 | 3601 | 3410 | 3475 |
| MIROC6 | 1038 | 989 | 984 | 966 | 3314 | 3273 | 3252 | 3225 |
| CMCC-ESM2 | 1185 | 1039 | 1046 | 1130 | 3441 | 3307 | 3348 | 3343 |





**Table A3.** Number of AR tracks detected in each dataset.

| | ARs tracks | | | |
|---|---|---|---|---|
| | Historical | SSP1-2.6 | SSP2-4.5 | SSP5-8.5 |
| ERA5 | 1224 | - | - | - |
| MPI-LR | 1283 | 1282 | 1223 | 1246 |
| MPI-HR | 1218 | 1251 | 1209 | 1198 |
| EC-Earth3 | 1186 | 1213 | 1222 | 1249 |
| NorESM2-MM | 1234 | 1261 | 1270 | 1247 |
| MIROC6 | 1151 | 1234 | 1200 | 1221 |
| CMCC-ESM2 | 1185 | 1203 | 1234 | 1207 |



## 335 A5 Cyclone intensity differences between future scenarios and historical simulations

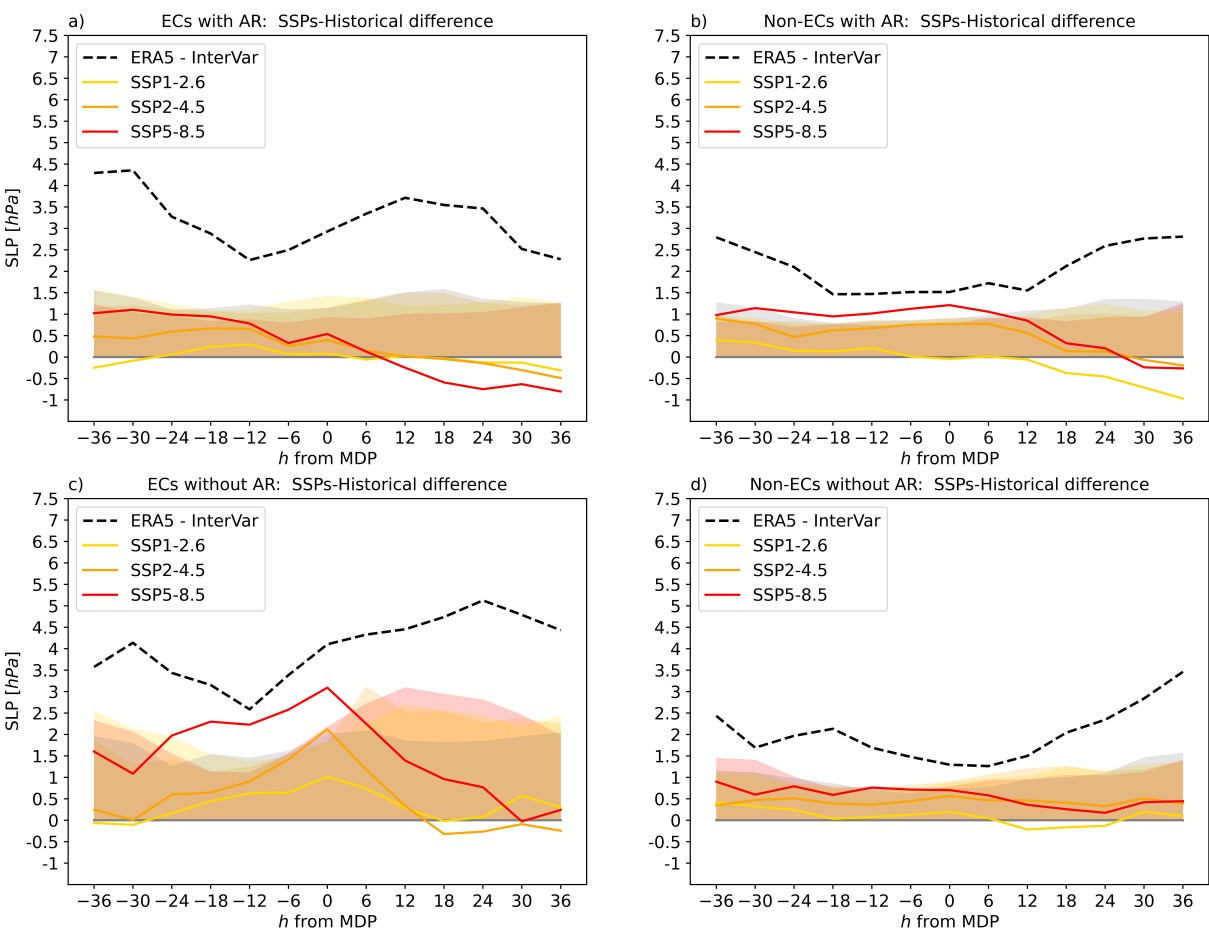

**Figure A1.** Difference in the multi-model mean SLP between the forcing scenarios and historical runs in solid lines, the inter-seasonal variability of the multi-model ensembles of mean SLP in shades and the inter-seasonal variability for ERA5 in the dashed line, for ECs with ARs (a), non-EC with ARs (b), ECs without ARs (c), and non-ECs without ARs (d).

*Author contributions.* FLM and MG developed the concept of the paper, performed the data analysis, prepared the figures and wrote the first manuscript draft. All authors contributed with ideas, interpretation of the results, and manuscript revisions.

*Competing interests.* Author GM is a member of the editorial board of Earth System Dynamics journal.



*Acknowledgements.* The authors wish to thank Lionel Guez, Atef Ben Nasser, Leonardo Olivetti, and Anastasiya Shyrokaya for useful dis-
cussions and technical support that shaped this article. This project has received funding from the European Union's Horizon 2020 research
and innovation programme under Marie Skłodowska-Curie grant No. 956396 (EDIPI project). GM further acknowledges the European
Union's H2020 research and innovation programme under ERC grant no. 948309 (CENÆ Project). DF and MG further acknowledge the
support of the COST Action FutureMed CA22162 supported by COST (European Cooperation in Science and Technology), an INSU-CNRS-
LEFE-MANU grant (project CROIRE), the state aid managed by the National Research Agency under France 2030 bearing the references
ANR-22-EXTR-0005 (TRACCS-PC4-EXTENDING project), the European Union's Horizon 2020 research and innovation programme un-
der grant agreement No. 101003469 (XAIDA).



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
