# Peer review of "Future Changes of Compound Explosive Cyclones and Atmospheric Rivers in the North Atlantic"

_EGUsphere, 2024_

## Author Comment (AC2)

This manuscript investigates the concurrences of explosive mid-latitude cyclones and atmospheric rivers and their future changes in the North Atlantic. The authors use ERA5 and CMIP6 climate models, tracking software, and study the link between ECs and ARs in the present and future climate. The main finding is that the concurrences of ECs and ARs is going to increase in the future, regardless that the number of ECs themselves show a decreasing trend.

I enjoyed reading the manuscript. I think it's very well written and easy to follow. The language was good, and the used datasets and methods are appropriate for the design of the study. Thanks to the authors for that. I also think that the topic of the study falls within the scope of the journal.

Despite the overall good presentation of the manuscript, I have some minor concerns related to the methods and the main result. I have listed them below, in addition to some line-to-line comments. I hope these are of help to the authors and they can address them before the paper is published.

First of all, we would like to thank the reviewer Mika Rantanen for the helpful comments on the manuscript.

1. Methods. At L134 you say that the ARs are detected within a 1500 km radius from the cyclone centre. At quick thinking, it sounds quite a large distance, given that e.g. Rudeva and Gulev (2007) found that the effective radius of oceanic ETCs is about 900 km. In the Introduction, you write (probably correctly) that the release of latent heat by the moisture of the ARs is an important mechanism in deepening the ECs. For me, it feels that if the AR is located very far (such as > 1000 km) from the cyclone centre, it cannot be involved in the deepening of the system. So, how did you arrive at the 1500 km value, and have you investigated how sensitive your results are to the used distance?

Thanks for this comment. The main reason to use the threshold of 1500 km for the radius detection of ARs in the surroundings of a cyclone is to be able to compare our results with the study of Eiras-Barca et al. (2018), where they evaluated the concurrence of ARs and cyclones with this radius for the historical period but using different tracking algorithms. In addition, we believe that the moisture brought by an AR (even if this is located further than 900 km) still influences the cyclone as in many cases is not the AR delivering its moisture directly to the cyclone centre but is the WCB or the feeder airstream that connect the enhanced moisture area of the AR with the cyclone and ultimately enhancing its intensification (Dacre et al. 2019). For this reason, ARs within 1500 km of the cyclone can contribute to its deepening as other airflows within the cyclone transport the moisture that potentially contributes to intensification. Finally, we tested the distribution of AR around the cyclone centre and detected a decline of ARs after 1500 km, this motivated us to keep the 1500 km already used by Eiras-Barca et al. (2018) and would keep our results comparable to theirs. Will provide this test and the sensitivity of the detection radius.

2. Results. Perhaps the headline result of your study is that the concurrences of ECs and ARs show an increasing trend in a warmer climate. In Sect 5.1 you

discuss that the number of individual ECs show a downward trend with climate change, and also that the change of detected ARs is very modest, almost flat, in a warming climate. At L201-204 you say that the increase of concurrence in a warmer climate points to the changes in characteristics or ARs or cyclones, i.e. as I understood, changes in the dynamics.

In any case, I was still missing a more detailed explanation or mechanism of how the concurrence of ECs and ARs can increase with climate change when neither individually shows a clear upward trend (I think this is a rather important finding which I haven't heard before!). I understand it may be challenging to find any clear explanation, but in the absence of one, it would be good to at least state out loud this dilemma clearer, for example in the conclusions or in the abstract. Now it feels like it is being swept under the rug as it is only briefly mentioned at L202-204 and not again in the conclusions.

Thanks for the comment, we agree that would be great to have a physical explanation of the results found here, and we hope this study encourages future research in this direction. Unfortunately, our results we cannot state which are the reasons for these changes but we will expand this in the discussion and add it to the conclusions.

Line-to-line comments:

L42. "Extratropical Cyclones". Do you mean explosive cyclones? If not, please decapitalize.

We mean extratropical cyclones, we will decapitalise.

L45. "undergoes further amplification compared to surface water vapor". I'm not really sure what you mean by further amplification and why this is the case. This sentence could be rephrased.

We mean that integrated water vapour (IWV) is expected to experience a larger increase than surface water vapour under climate change. We will rephrase the sentence to make it clearer.

L73. Why did you select 2009 as the ending year? Was that because you wanted to have 30 years in the historical period, consistently with 2070-2099 for the future? Please add an explanation to the text. And also, does your time range include OND 1979, i.e. should it be ONDJFM 1979/1980 - 2008/2009?

We used until 2009 for the historical period to have 30 years of data [1980-2009], the same as for the future periods [2070-2099]. The data includes 30 "natural" years and from there we subtract the winter months, in total 180 months of data each period. This means that each period has 29 full winters (ONDJFM) and two half winters, JFM for the first year and OND for the last year.

L95. What's the unit of NDR? Isn't it hPa / h. Now there's no unit after 1.

The NDR does not have units, it is a dimensionless variable (Lim and Simmonds, 2002). Apologies for the confusion, formula (1) in the denominator of the first fraction should be 24 hPa (not 24 h). The DR_24h unit is also in hPa, thus when divided by 24 hPa the result is the dimensionless index NDR.

L99. Here you start to speak about the number of cyclones detected. However, I think you should more clearly repeat the domain of tracking. The domain of ERA5 was presented at L74. Is this the same domain where you apply the tracking software?

The domain where the tracking is applied is the same for all datasets [25-65ºN; 80ºW-10ºE], now we will apply a buffer zone of 10º at each boundary of this domain (ERA5 and all CMIP6 models). Thanks for this comment, we will be more clear with the tracking domain in the manuscript. For further explanation of the tracking area and the implementation of the buffer zone, I kindly refer to our answer to the second comments of Referee 2 for a more detailed explanation.

L99. How do you treat those cyclones which form or decay outside the domain and only travel across it? How can you be sure what the MDP of a given cyclone is if only part of the cyclone's life cycle occurs inside the domain?

First, cyclones can only have their tracks inside of the domain [25-65ºN; 80ºW-10ºE], and the MDP is calculated with these tracks so the MDP corresponds to the part of the track inside the domain. If a cyclone has the MDP before/after entering/leaving the domain we could not detect that MDP. Now, with the implementation of the +10º buffer zone at the boundaries, the tracking is applied to a larger domain, and the MDP is calculated for the tracks in the extended domain with the buffer area. Then, the analysis only includes the parts of the tracks in the original domain [25-65ºN; 80ºW-10ºE], thus MDP can be outside of the domain but the analysis is only done with the part of the track inside. This ensures that we detect the correct MDP, and improves the representation of the cyclones that form or decay outside the domain.

L117-118. These threshold values seem a bit subjective. How did you arrive at them? I think the justification for these values should be mentioned.

These thresholds are defined and tested by the TempestExtrem developers, we use the same as in Ullrich et al. (2021). In addition, we add the threshold most used in AR detection and tracking algorithms (IVT>250kg/m/s) for validation. The TempestExtrem tracking algorithm used in our study has been compared to other tracking algorithms showing a large agreement (Collow et al. 2022). Thanks again, we will add this to the manuscript.

L118. "The detected candidates are concatenated if at least one grid point is detected as AR in sequential timesteps". Does this mean that the AR area at the next time step must overlap spatially with the AR area at the previous time step?

Exactly, we will improve the sentence to make it more clear.

L134. The presence of AR. I understand that the location of the EC is clearly defined, i.e. it means the location of the minimum SLP. Yes. But it is unclear to me how the location of the AR is defined in relation to the EC. If the AR is wide, does that mean that it is sufficient if the closest grid point of the AR is at most 1500 km from the centre of the cyclone?

Yes, or in other words: if at least one grid point detected as an AR is within 1500 km from the centre of the cyclone, then that cyclone is concurrent with an AR. We will improve this sentence to make this clear.

Fig. 3. Does the map show Xynthia's full life cycle, the whole track? Not a big deal, but I missed the MDP of Xynthia. Can you show its location on the map? I think it would better tie Fig. 3 to the following figures where the time axis is shown in relation to MDP.

Yes, it shows Xynthia's full track (the track we detect with our algorithm). Thanks for the recommendations, we'll include the MDP point and time with respect to the MDP.

L146-147. "Initial stages of the cyclone formation / dissipation stages of the cyclones". In Fig. 4, you show only 72 hours (3 days) of the cyclone composites. Arguably many of the tracked cyclones last longer, meaning that their formation or decay can be days before/after MDP. Figure 7 shows that, on average, the minimum SLP of the cyclones has not increased much even 36 hours after the MDP. So is it correct to speak of dissipation in this context? Could it be better to just say 36 hours after / before MDP?

Thanks for this comment, other reviewers also pointed this out. We agree that formation/dissipation stages might not be appropriate as we analyse the cyclones +/-36 hours from the MDP. We will change this and modify the text accordingly to avoid referring to formation/dissipation stages in this context.

L157 and hereafter. For me, inter-seasonal means variations or comparisons between different seasons within a single year. So basically changes and differences from one season to another within the same year. Whereas inter-annual refers to variations between the same periods in different years, i.e. year-to-year differences. Do you mean inter-annual here and later in the manuscript?

Thanks again for this comment, other reviewers also ask the same. We refer to variations between the same periods in different years (we only analyse extended winter). We will change inter-seasonal for inter-annual in all the manuscript.

L174. What does the internal variability of the datasets mean in this context? I think the sentence needs rephrasing.

It means inter-annual variability, we will correct that.

L177. quantitative We will fix the typo, thanks.

Fig. 4. Could it be written to the panels the a-b represent averages over 30 years, and c-d standard deviations? It took me a while to understand that the panels c-d are

standard deviations, especially as I got confused about the meaning of "inter-seasonal" in their title (see the comment a few comments back).

Thanks for the comment. We will improve the clarity of Figure 4 regarding this, and we will make sure that we use inter-annual variability throughout the manuscript and make sure it is well explained what it means.

L184 and so on: 12 % or 12 percentage points?

12 percentage points. We will re-write this accordingly.

L192. "for almost all the models". Well that's one way of saying it if 4 out of 6 models are showing an increase of AR tracks between SSP5-8.5 and historical. In general, the values in Table A3 seem very unchanged, so I think it could be honest to say that there is not really a systematic change at all between the scenarios and historical. It could be just random variation (internal climate variability).

Thanks for the comment,

L199. Does this sentence refer to Fig. 5a? The reference to the figure could be added.

Yes, we will refer to Fig. 5a in the text.

L200. By the number of compound events do you mean a situation where at least one time step of the track of the cyclone centre is closer than 1500 km to the AR?

We mean that the total number of time steps that a cyclone (EC or non-EC) is concurrent with an AR increases. Before and in Fig.5 we show an increase in the rate of coincidence (increase in the ratio), here we want to emphasise that the total number (or absolute number) also increases. Because an increase in the ratio could be due to a decrease in the total number of cyclone time-steps, but not necessarily an increase of concurrent cyclone/AR time-steps.

L207. the inter-model spread of the SSP5-8.5 scenario Will correct this.

L228. CMIP6 Will correct the typo. Thanks.

L249. "... is very limited". I think you could continue this sentence by for example "as the coloured lines in Fig. 7 are close to each other" or similar. It took me some time to understand where you got this conclusion.

We will expand this sentence and refer to Fig. 7 for better clarity.

Fig. 7e. I think you do not discuss at all why CMIP6 models seem to be more sensitive to the ARs than ERA5? Or did I understand it correctly? Why the coloured lines in Fig. 7e go much lower before MDP and much higher after MDP when compared to ERA5?

We will add this in the discussion of the results of Figure 7.

Conclusions. Currently, I think the conclusions (and in fact the whole paper) paper puts quite a lot of emphasis on the high emission SSP5-8.5 scenario. However, it has been shown to be unrealistic (https://www.nature.com/articles/d41586-020-00177-3), and the world is currently roughly on the path of the SSP2-4.5 scenario. It might be appropriate to add a few sentences of discussion on this, stating that the results should be interpreted always with the scenario in mind, and that the results of the SSP2-4.5 scenario are more likely in the future than those of SSP5-8.5.

Thanks for this point, actually our motivation to analyse 3 different emissions scenarios was that some of those can be unrealistic, thus we believe is more important to study more than one scenario. We will add this in the text and will make sure that results for all scenarios are discussed.

L283. SSP5-8.5 scenario Will correct the typo. Thanks.

Table A2-A3. It could be helpful to add the periods (the year ranges) used for historial and SSP scenarios also here.

We will do it, I hope it will help to interpret the table.

References

Rudeva and Gulev (2007): https://journals.ametsoc.org/view/journals/mwre/135/7/mwr3420.1.xml

Additional References:

Eiras-Barca et al. (2018): https://esd.copernicus.org/articles/9/91/2018/

Dacre et al. (2019): https://journals.ametsoc.org/view/journals/hydr/20/6/jhm-d-18-0175_1.xml

Zhang et al. (2018): https://agupubs.onlinelibrary.wiley.com/doi/full/10.1029/2018GL079071

Lim and Simmonds (2002): https://journals.ametsoc.org/view/journals/mwre/130/9/1520-0493_2002_130_2188_ecdits_2.0.co_2.xml

Ullrich et al. (2021): https://gmd.copernicus.org/articles/14/5023/2021/

McClennyet al. (2021): https://agupubs.onlinelibrary.wiley.com/doi/10.1029/2020JD033421

Collow et al. (2022): https://agupubs.onlinelibrary.wiley.com/doi/full/10.1029/2021JD036155

---

## Author Comment (AC3)

This manuscript aims to evaluate the current climatology and assess changes under future climate scenarios of the concurrence between atmospheric rivers and extratropical cyclones undergoing explosive development – frequently referenced as explosive cyclones – in the North Atlantic. Being the explosive development of extratropical cyclones and atmospheric rivers crucial in driving extreme weather in the mid-latitudes, this topic is relevant, deserves to be investigated and it fits the scope of the Earth System Dynamics journal.

The manuscript is well-structured and well written and it is pleasant to read. It applies well-known datasets and detection and tracking methods previously published and discussed in the literature. However, in my opinion, some points are too succinct and need further details and explanations before the manuscript is accepted for publication.

First of all, we would like to thank the anonymous reviewer for the helpful comments on the manuscript.

Some points that need further clarification are:

- The authors use TempestExtremes Code for Detecting and Tracking Extratropical Cyclones and Atmospheric Rivers (ARs) for the North Atlantic region [25-65ºN; 80◦W-10◦E]. In Appendix "A4 Number of Cyclones and ARs detected in ERA5 and CMIP6" the words Cyclone and track are used as synonyms; in Figure 1 "EC track density climatology" is presented and "Units are the number of cyclones per 1.5◦ spherical cap per month"; and in Figure 2 "AR frequency climatology" is presented with "Units are the percentage of time-steps detected as AR". Before a climatology can be presented and discussed, a clarification must be presented, and methods should detail the definitions and how this has been computed. In both cases, the authors must clarify if the systems are being tracked and considered or if the systems' timesteps are considered independently. All the processes to produce the cyclones' tracks and ARs datasets must be better explained. From my understanding, the systems are considered, but this is not clear from the discussion and captions of Figures 1, 2 and 8.

This is correct (the systems are being tracked and considered), we agreed the methodology needs clarification. We will ensure that the text clearly states what is being referred to in each case and will harmonise the nomenclature for consistency. Additionally, we will expand the Methods section to describe how the climatologies presented in Figures 1 and 2 are calculated. In the main body of the manuscript (including Figures 1 and 2), we refer to the time steps of the tracked systems (the systems' time steps are considered independently). Only in Appendix A4 do we refer to the total number of tracked systems (the systems are being tracked and considered).

- The North Atlantic region [25-65ºN; 80◦W-10◦E] is considered. The authors should discuss the artefact over the western boundaries of the domain. A buffer area should be considered for the identification and tracking of the systems.

Thank you for highlighting this issue. We acknowledge the artefact in the western boundary of the domain in cyclone climatology. We identified that the cyclone tracking algorithm creates this issue specifically at the western boundary, where cyclones move eastward as they enter the domain. It creates stationary "artefact" cyclones that have

their MDP along the boundary. To address this and ensure it does not affect our results, we will apply a 10º buffer zone at all boundaries of the domain. The new tracking domain will be 15-75ºN, 90ºW-20ºE, while for the analysis, only the time steps of tracks within the original domain [25-65ºN, 80ºW-10ºE] will be considered.

We found that this issue was impacting the concurrence results in Figures 4 and 5, where the peak of concurrence was initially 6 hours after the MDP. By correcting this issue and adding the buffer zone, the peak of concurrence now aligns with the MDP. This correction brings our results in line with those of Eiras-Barca et al. (2018). The previous shift in the peak of concurrence was due to the "artefact" cyclones having the MDP at the boundary meaning that these cyclones had the MDP at the first time steps of their tracks. This shifted the curves in Figures 4 and 5 to the right as the "artefact" cyclones were adding a bias only to the times after the MDP. All results that depend on the tracking will be updated.

- The method to identify the concurrences of Extratropical Cyclones and ARs also needs further explanation: it is presented through Figure 3 and the Xynthia case study, but this example elucidates the doubts and need for clarification on the methods. From this example, five timesteps are consistent with the concurrence of the cyclone under explosive development and the occurrence of the AR. The shaded areas in Figure 3 that depict the regions identified as ARs should have some correspondence with the cyclone track and should be described in the text as well. It is not clear to me how many times cyclone Xynthia and the concurrent AR are considered for the climatological assessment. I would say Figures 1 and 2 correspond to timesteps – and not Cyclones/AR.

We will improve the description of the methodology to make it as clear as possible. We will also modify Figure 3 (see also reply to Mika Rantanen) to make it more useful to understand the methodology. As you suggest, we will include in Figure 3 which AR (shades) correspond to which time step in the cyclone track (crosses). Each cyclone time step and the concurrent AR (or not) is only used once in the climatology assessment or further in the following results sections (same for Xynthia).

- As mentioned previously, the method to identify the concurrences (Ln 130-134) must be further detailed. Please discuss the choice of the Maximum Deepening Point (MDP). An explanation should be given for the choice of the 1500 km threshold. It is not clear if a sensitivity analysis was performed, nor if this metric is constant for all cyclone's sequential timesteps. To the best of my understanding, each detected AR candidate may have more than one grid point being detected as AR in the same timestep. Certainly, the authors have considered this and all these aspects should be presented and discussed in the methods section. Additionally, how this method differs from Eiras-Barca et al. (2018) should be highlighted.

Thank you once again for your insightful comment. We will revise the text in the Methods section for clarity. We selected the maximum deepening point (MDP) as a time reference because it allows us to better assess the influence of ARs on cyclone cyclogenesis. Additionally, using the MDP makes our results directly comparable to those of Eiras-Barca et al. (2018).

Regarding the choice of the 1500 km threshold, I kindly refer you to our response to the first comment in our reply to Mika Rantanen. As you noted, our methodology aims to align as closely as possible with that of Eiras-Barca et al. (2018) to ensure comparability. The main difference lies in the tracking methodologies for ARs and cyclones, as we use different tracking algorithms but still, those algorithms share similar configuration parameters. However, in terms of cyclone/AR concurrence detection, explosive versus non-explosive cyclone classification, and the calculation of the MDP as a time frame, we follow the same methodology. We will emphasize all these points in the Methods section.

- The clarification of these methodological aspects is vital for the discussion of the results: authors should clearly state if this study evaluates "the concurrence of ECs and ARs in the ERA5 reanalyses and we compare it with those obtained in climate models" considering only once each EC and AR.

We will emphasize in the text that we only consider once each cyclone (EC or non-EC) and AR time step in our analysis.

- All the analysis is performed for the extended winter period (October to March). This should be indicated in the figure captions. Please clarify, in the methodology, how the inter-seasonal variability is defined if only the extended winter season is considered. I suppose the authors mean interannual variability.

Thanks again for helping us to explain better the methodology. We will add in Figure captions that the period analyzed is extended winter (ONDJFM).

We mean interannual variability (variations between extended winters in different years), we will change interseasonal for interannual everywhere in the manuscript. The interannual variability is defined as the standard deviation among the 30 years of each period, we will add this definition in the methods section.

- CMIP6 models' information and discussion of results: additional detail should be included for the choice of one single member for each model and not the ensemble – how this particular member has been selected and how this choice may affect the final results. This should be included in the methodology and the discussion. Please refer to whether one may state that model X overestimates/underestimates the results or if model Y is more adequate for the analysis if only one member has been used. Please also discuss if the biases quantification is reliable. To the best of my knowledge, this assessment is not enough to make a comparison between models. A multimodel ensemble framework with varied combinations of GCMs is extremely useful and allows for reducing the uncertainty in climate projections for future scenarios and for a tendency assessment, but it can hardly be used to intercompare models when only one member is used. Please, define "the internal variability of the datasets" (ln 161) in the methodology section and how it is assessed in this manuscript.

The limitation on the number of CMIP6 models and ensemble members is explained in lines 80-82. We will further elaborate on this in the Methods section and include it in the Discussion and Conclusions. We were restricted to using one member per model because the other ensemble members did not have the necessary variables to calculate IVT, which is essential for studying ARs. In essence, we used all available members from

CMIP6 models that had the required variables for the historical period and the three scenarios.

We acknowledge, both here and in the manuscript, that the limited number of members is a limitation. For this reason, we assess changes between the present and future using the multi-model mean of the ensemble (ensemble of 6 members from 6 different models). We also agree that stating a particular model overestimates or underestimates results may not be appropriate when using only one member per model, and we will revise this wording in the text.

In line 161, we refer to "the internal variability of the datasets" as the model's spread, or in other words, the spread within the multi-model ensemble. We will clarify this in the manuscript.

- It would be useful if the results presented in the Appendix should be accompanied by a short description and discussion. Please avoid using expressions like "little change" (ln 190-191) or "lower-than-expected increase" (ln 194)– please quantify. A percentage could be added to tables.

Thanks for the comment, we will add a description and improve the information in the Appendix. We will review the text to avoid these expressions.

- Ln 200 – this sentence deserves additional information or a reference. It is out of context in this paragraph. These would be relevant results but evidence must be shown.

We will provide additional context for this sentence. For a more detailed explanation, I kindly refer you to our response to Mika Rantanen.

- Conclusions: please discuss what is the novelty, for the present period, from the literature. The sentence "The fact that AR concurrences are larger after the MDP suggests that mature ECs (when they are deeper) can facilitate the formation of ARs in their surroundings" (ln 272-273) deserves to be further discussed and justified. Firstly, it is well known that the detecting and tracking methods still have large uncertainty in detecting the absolute minimum central pressure of an extratropical cyclone; secondly, the difference should be quantified; finally, and most importantly, the only conclusion that these results allow us to obtain, in this state, is that additional AR are detected – we cannot state that they only formed at that particular timestep.

Thank you for your comments. We will ensure the novelty of our results is emphasized. In lines 272-273, we intend to convey that as more ARs are concurrent with ECs around the MDP, this suggests that ARs are more likely to occur when the EC is at its maximum deepening stage. This finding is supported by other studies, such as Zhang et al. (2018) and Eiras-Barca et al. (2018). This result holds true for both the historical period and all future scenarios. Additionally, we demonstrate that concurrences increase across future scenarios at all stages, not just around the MDP.

We agree and acknowledge that tracking algorithms come with inherent uncertainties. We have quantified the number of individual cyclone tracks detected (Table A2 in Appendix A4) and will highlight and expand these results in the text. As mentioned in a

previous comment, we will also include the percentage changes in the table. While our results have not identified clear changes in cyclone track counts, we believe that despite uncertainties introduced by tracking algorithms, our results (among other findings) confidently indicate an increase in the coincidence between cyclones and ARs, as well as an increase in ARs and their intensity.

References:

Eiras-Barca et al. (2018): https://esd.copernicus.org/articles/9/91/2018/

Zhang et al. (2018): https://agupubs.onlinelibrary.wiley.com/doi/full/10.1029/2018GL079071

---

## Author Response (AR2)

**Future Changes of Compound Explosive Cyclones and Atmospheric Rivers in the North Atlantic**

Submitted to Earth System Dynamics

**Authors' response**

We would like to thank the four reviewers for their helpful comments on the manuscript. In the following, the referee's comments are in black and our answers are below each of them in red We further indicate the most relevant adjustments made to the manuscript in **bold red** below our answers where applicable. All line numbers refer to the revised manuscript.

**1. Response to comment by Mika Rantanen**

This manuscript investigates the concurrences of explosive mid-latitude cyclones and atmospheric rivers and their future changes in the North Atlantic. The authors use ERA5 and CMIP6 climate models, tracking software, and study the link between ECs and ARs in the present and future climate. The main finding is that the concurrences of ECs and ARs is going to increase in the future, regardless that the number of ECs themselves show a decreasing trend.

I enjoyed reading the manuscript. I think it's very well written and easy to follow. The language was good, and the used datasets and methods are appropriate for the design of the study. Thanks to the authors for that. I also think that the topic of the study falls within the scope of the journal.

Despite the overall good presentation of the manuscript, I have some minor concerns related to the methods and the main result. I have listed them below, in addition to some line-to-line comments. I hope these are of help to the authors and they can address them before the paper is published.

First, we would like to thank the reviewer, Mika Rantanen, for the valuable comments on our manuscript, which have greatly contributed to its improvement.

1. Methods. At L134 you say that the ARs are detected within a 1500 km radius from the cyclone centre. At quick thinking, it sounds quite a large distance, given that e.g. Rudeva and Gulev (2007) found that the effective radius of oceanic ETCs is about 900 km. In the Introduction, you write (probably correctly) that the release of latent heat by the moisture of the ARs is an important mechanism in deepening the ECs. For me, it feels that if the AR is located very far (such as > 1000 km) from the cyclone centre, it cannot be involved in the deepening of the system. So, how did you arrive at the 1500 km value, and have you investigated how sensitive your results are to the used distance?

Thanks for this comment. The main reason to use the threshold of 1500 km for the radius detection of ARs in the surroundings of a cyclone is to be able to compare our results with the study of Eiras-Barca et al. (2018), where they evaluated the concurrence of ARs and cyclones with this radius for the historical period but using different tracking algorithms. In addition, we believe that the moisture brought by an AR (even if this is located further than 900 km) still

influences the cyclone as in many cases is not the AR delivering its moisture directly to the cyclone centre but is the WCB or the feeder airstream that connects the enhanced moisture area of the AR with the cyclone and ultimately enhancing its intensification (Dacre et al. 2019). For this reason, ARs within 1500 km of the cyclone can contribute to its deepening as other airflows within the cyclone transport the moisture that potentially contributes to intensification.

We have analyzed the distribution of the closest points of atmospheric rivers (AR) around the cyclone center (Fig. R1, and Fig. S1 of the supplementary material) and found that most of them fall within the east-southeast quadrant of the 1500 km circle. This is expected, as most ARs that are dynamically associated with a cyclone are located to the southeast of the cyclone center, linked to the WCB and feeder airstream. This pattern is further illustrated in Figure A2, where the histogram shows that most ARs are in the southeast quadrant. If we expand this radius to 2500 km, there is a shift in AR occurrences towards the southwest and northwest quadrants of the cyclone (Fig. R2, and Fig. S2 of the supplementary material), making it less likely that these ARs are dynamically linked to the cyclone. Therefore, we conclude that 1500 km represents a good compromise between maximizing the number of possible cases (statistics) and selecting those that are dynamically linked (dynamics).

[Figure]

*Figure R1. Closest atmospheric river (AR) points found around cyclones, both detected using ERA5 data. Points in red indicate those found within a 1500 km circle from the cyclone center, and in gray those found up to 2500 km away.*

[Figure]

*Figure R2. Percentage of occurrence (left) of the cyclone quadrant in which the closest atmospheric river (AR) was found within a 1500 km circle (brown) and a 2500 km circle (gray). The right figure shows the probability distribution function (PDF) of the distance, in kilometers, between the cyclone center and the closest point of the AR.*

We have included these two figures in the Supplementary material (Fig. S1 and S2), as well as included the following justification in the article:

- Lines 149–156: "Subsequently, we determine whether a specific timestep of an extratropical cyclone (EC or non-EC) is linked to an AR by detecting at least one grid point classified as an AR within 1500 km from the centre of the cyclone. **Hence, each detected cyclone may have more than one grid point detected as an AR. This 1500 km radius is consistently applied across all time steps of the cyclone tracks. By selecting a 1500 km radius, our methods align with those of (Eiras-Barcas et al. 2018), with the primary difference between the two methods being the AR and cyclone tracking algorithms used. We consider that moisture brought by an AR may influence the cyclone within this radius by delivering moisture to the warm conveyor belt (WCB) or feeder airstream (Dacre et al. 2019). Most of the identified ARs are located in the southeastern quadrant of the cyclone (Supplementary Figures S1 and S2), which maximizes the probability that the AR and cyclone are dynamically linked through these two components.**"

2. Results. Perhaps the headline result of your study is that the concurrences of ECs and ARs show an increasing trend in a warmer climate. In Sect 5.1 you discuss that the number of individual ECs show a downward trend with climate change, and also that the change of detected ARs is very modest, almost flat, in a warming climate. At L201-204 you say that the increase of concurrence in a warmer climate points to the changes in characteristics or ARs or cyclones, i.e. as I understood, changes in the dynamics.

In any case, I was still missing a more detailed explanation or mechanism of how the concurrence of ECs and ARs can increase with climate change when neither individually shows a clear upward trend (I think this is a rather important finding which I haven't heard before!). I understand it may be challenging to find any clear

explanation, but in the absence of one, it would be good to at least state out loud this dilemma clearer, for example in the conclusions or in the abstract. Now it feels like it is being swept under the rug as it is only briefly mentioned at L202-204 and not again in the conclusions.

Thanks for the comment, we agree that would be great to have a physical explanation of the results found here, and we hope this study encourages future research in this direction. Unfortunately, our results can not state which are the reasons for these changes but we will expand this in the discussion and add it to the conclusions:

- Lines 341–346 (Conclusions): "Our results show a generalized increase in compound events of ECs with ARs in most of the North Atlantic basin. A robust increase in concurrence over the British Isles, Iberia and north France is only observed under the most severe climate change scenario. **However, we did not detect a clear upward trend in the individual frequency of ECs or ARs across the entire North Atlantic basin. This apparent contradiction suggests that changes in the characteristics or dynamics of ECs and ARs, rather than their frequency, may be driving the observed increase in concurrence. This is a significant finding that needs further investigation, as the underlying physical mechanisms for this increase remain unclear."**

Line-to-line comments:

L42. "Extratropical Cyclones". Do you mean explosive cyclones? If not, please decapitalize.

We mean extratropical cyclones, we have decapitalised.

L45. "undergoes further amplification compared to surface water vapor". I'm not really sure what you mean by further amplification and why this is the case. This sentence could be rephrased.

We mean that integrated water vapour (IWV) is expected to experience a larger increase than surface water vapour under climate change. We rephrased the sentence to make it clearer:

"This increase is driven by the Clausius-Clapeyron relation, which implies a rise in moisture content in a warmer atmosphere. **However, integrated water vapour (IWV) is expected to experience a larger increase than surface water vapour under climate change.**"

L73. Why did you select 2009 as the ending year? Was that because you wanted to have 30 years in the historical period, consistently with 2070-2099 for the future? Please add an explanation to the text. And also, does your time range include OND 1979, i.e. should it be ONDJFM 1979/1980 - 2008/2009?

We used until 2009 for the historical period to have 30 years of data [1980-2009], the same as for the future periods [2070-2099]. The data includes 30 "natural" years and from there we subtract the winter months, in total 180 months of data each period. This means that each period has 29 full winters (ONDJFM) and two half winters, JFM for the first year and OND for the last year. We have included it in the text:

- Lines 84–86: These 30-year datasets consist of 29 full winters and two partial winters (January to March for 1980 and 2070, and October to December for 2009 and 2099).

L95. What's the unit of NDR? Isn't it hPa / h. Now there's no unit after 1.

The NDR does not have units, it is a dimensionless variable (Lim and Simmonds, 2002). Apologies for the confusion, formula (1) in the denominator of the first fraction should be 24 hPa (not 24 h). The DR_24h unit is also in hPa, thus when divided by 24 hPa the result is the dimensionless index NDR. We have corrected equation 1.

L99. Here you start to speak about the number of cyclones detected. However, I think you should more clearly repeat the domain of tracking. The domain of ERA5 was presented at L74. Is this the same domain where you apply the tracking software?

The domain where the tracking is applied is the same for all datasets [25-65ºN; 80ºW-10ºE], now we will apply a buffer zone of 10º at each boundary of this domain (ERA5 and all CMIP6 models). Thanks for this comment, we will be more clear with the tracking domain in the manuscript. For further explanation of the tracking area and the implementation of the buffer zone, I kindly refer to our answer to the second comments of Referee 2 for a more detailed explanation.

We have added in Lines 76–79: " **(hereafter referred to as the Data Domain). To mitigate issues caused by the cyclone tracking algorithm, which tends to generate stationary 'artefact' cyclones along the western boundary, we apply a buffer zone of 10d° on this data domain. This adjustment ensures that artefacts at the western boundaries are excluded. As a result, the domain used for all analyses presented in the results section is [25–65°N; 80°W–10°E], referred to simply as domain from here on.**" We hope that this also clarifies the point raised by the reviewer.

L99. How do you treat those cyclones which form or decay outside the domain and only travel across it? How can you be sure what the MDP of a given cyclone is if only part of the cyclone's life cycle occurs inside the domain?

First, cyclones can only have their tracks inside of the domain [25-65ºN; 80ºW-10ºE], and the MDP is calculated with these tracks so the MDP corresponds to the part of the track inside the domain. If a cyclone has the MDP before/after entering/leaving the domain we could not detect that MDP. Now, with the implementation of the +10º buffer zone at the boundaries, the tracking is applied to a larger domain, and the MDP is calculated for the tracks in the extended domain with the buffer area. Then, the analysis only includes the parts of the tracks in the original domain [25-65ºN; 80ºW-10ºE], thus MDP can be outside of the domain but the analysis is only done with the part of the track inside. This ensures that we detect the correct MDP, and improves the representation of the cyclones that form or decay outside the domain.

L117-118. These threshold values seem a bit subjective. How did you arrive at them? I think the justification for these values should be mentioned.

These thresholds are defined and tested by the TempestExtrem developers, we use the same as in Ullrich et al. (2021). In addition, we add the threshold most used in AR detection and tracking algorithms (IVT>250kg/m/s) for validation. The TempestExtrem tracking algorithm

used in our study has been compared to other tracking algorithms showing a large agreement (Collow et al. 2022). Thanks again, we have added this to the manuscript:

- Lines 130–131: "These thresholds were defined and tested by Ullrich et al. 2021, and show strong agreement with other tracking algorithms (Collow et al. 2022)."

L118. "The detected candidates are concatenated if at least one grid point is detected as AR in sequential timesteps". Does this mean that the AR area at the next time step must overlap spatially with the AR area at the previous time step?

Exactly, we have improved the sentence to make it more clear.

- Line 132–133: "The detected candidates are concatenated if at least one grid point is identified as an AR in consecutive timesteps, **meaning that the AR area at a consecutive timestep spatially overlaps with the previous AR."**

L134. The presence of AR. I understand that the location of the EC is clearly defined, i.e. it means the location of the minimum SLP. But it is unclear to me how the location of the AR is defined in relation to the EC. If the AR is wide, does that mean that it is sufficient if the closest grid point of the AR is at most 1500 km from the centre of the cyclone?

Yes, or in other words: if at least one grid point detected as an AR is within 1500 km from the centre of the cyclone, then that cyclone is concurrent with an AR. We will improve this sentence to make this clear.

- Lines 149–151: "**Subsequently, we determine whether a specific timestep of an extratropical cyclone (EC or non-EC) is linked to an AR if at least one grid point within 1500 km from the centre of the cyclone is part of an AR (detected and tracked independently, see Section 3.2).**"

Fig. 3. Does the map show Xynthia's full life cycle, the whole track? Not a big deal, but I missed the MDP of Xynthia. Can you show its location on the map? I think it would better tie Fig. 3 to the following figures where the time axis is shown in relation to MDP.

Yes, it shows Xynthia's full track (the track we detect with our algorithm). Thanks for the recommendations, we have included the MDP point and time with respect to the MDP:

[Figure]

L146-147. "Initial stages of the cyclone formation / dissipation stages of the cyclones". In Fig. 4, you show only 72 hours (3 days) of the cyclone composites. Arguably many of the tracked cyclones last longer, meaning that their formation or decay can be days before/after MDP. Figure 7 shows that, on average, the minimum SLP of the cyclones has not increased much even 36 hours after the MDP. So is it correct to speak of dissipation in this context? Could it be better to just say 36 hours after / before MDP?

Thanks for this comment, other reviewers also pointed this out. We agree that formation/dissipation stages might not be appropriate as we analyse the cyclones +/-36 hours from the MDP. We have changed this and modified the text accordingly to avoid referring to formation/dissipation stages in this context.

L157 and hereafter. For me, inter-seasonal means variations or comparisons between different seasons within a single year. So basically changes and differences from one season to another within the same year. Whereas inter-annual refers to variations between the same periods in different years, i.e. year-to-year differences. Do you mean inter-annual here and later in the manuscript?

Thanks again for this comment, other reviewers also ask the same. We refer to variations between the same periods in different years (we only analyse extended winter). We have changed inter-seasonal for inter-annual in all the manuscript.

L174. What does the internal variability of the datasets mean in this context? I think the sentence needs rephrasing.

It means inter-annual variability, we have corrected that.

L177. quantitative We have corrected the typo, thanks.

Fig. 4. Could it be written to the panels the a-b represent averages over 30 years, and c-d standard deviations? It took me a while to understand that the panels c-d are standard deviations, especially as I got confused about the meaning of "inter-seasonal" in their title (see the comment a few comments back).

Thanks for the comment. We have used inter-annual variability throughout the manuscript and made sure it is well explained what it means.

L184 and so on: 12 % or 12 percentage points?

12 percentage points. We have re-written this accordingly.

L192. "for almost all the models". Well that's one way of saying it if 4 out of 6 models are showing an increase of AR tracks between SSP5-8.5 and historical. In general, the values in Table A3 seem very unchanged, so I think it could be honest to say that there is not really a systematic change at all between the scenarios and historical. It could be just random variation (internal climate variability).

Thanks for the comment, we have rephrased it.

L199. Does this sentence refer to Fig. 5a? The reference to the figure could be added.

Thank you, we have referred to Fig. 5a in the text.

L200. By the number of compound events do you mean a situation where at least one time step of the track of the cyclone centre is closer than 1500 km to the AR?

We mean that the total number of time steps that a cyclone (EC or non-EC) is concurrent with an AR increases. Before and in Fig.5 we show an increase in the rate of coincidence (increase in the ratio), here we want to emphasise that the total number (or absolute number) also increases. Because an increase in the ratio could be due to a decrease in the total number of cyclone time-steps, but not necessarily an increase of concurrent cyclone/AR time-steps.

We have rephrased it as:

- Lines 227–234 : "**The increase in the rate of coincidence between cyclones and ARs could partly result from a decrease in the total number of cyclone time-steps, rather than a direct increase in the number of concurrent cyclone/AR time-steps. To clarify this, we also calculated the absolute number of ECs and non-ECs concurrent with ARs (not shown) and found that this number increases across all scenarios. Hence, the combined effect of a slight increase in AR frequency and a slight decrease in the number of ECs and non-ECs could, at least partially, explain the rise in the rate of coincidence as the level of warming increases (Figure 6a,b). These findings suggest that changes in the characteristics of ARs, cyclones, or their interactions may be driving the observed changes, rather than the result being merely a statistical artifact of more cyclones and ARs occurring individually.**"

L207. the inter-model spread of the SSP5-8.5 scenario

We have corrected this.

L228. CMIP6 We have corrected the typo. Thanks.

L249. "... is very limited". I think you could continue this sentence by for example "as the coloured lines in Fig. 7 are close to each other" or similar. It took me some time to understand where you got this conclusion.

We have expanded this sentence and referred to Fig. 7 for better clarity.

- Lines 184–186 : " The influence of climate change on cyclone intensity for any of the four types of compound events is very limited, **as the historical and scenario lines, along with their respective spreads in Fig. 7, are close to each other or overlap**."

Fig. 7e. I think you do not discuss at all why CMIP6 models seem to be more sensitive to the ARs than ERA5? Or did I understand it correctly? Why the coloured lines in Fig. 7e go much lower before MDP and much higher after MDP when compared to ERA5?

We have made significant changes to Figure 7 to reflect the revised cyclone tracking following the application of the buffer zone. The new results from ERA5 show similar behavior for both types of cyclones when compared to the models, and these findings have been incorporated into the manuscript.

Conclusions. Currently, I think the conclusions (and in fact the whole paper) paper puts quite a lot of emphasis on the high emission SSP5-8.5 scenario. However, it has been shown to be unrealistic (https://www.nature.com/articles/d41586-020-00177-3), and the world is currently roughly on the path of the SSP2-4.5 scenario. It might be appropriate to add a few sentences of discussion on this, stating that the results should be interpreted always with the scenario in mind, and that the results of the SSP2-4.5 scenario are more likely in the future than those of SSP5-8.5.

Thanks for this point, actually our motivation to analyse 3 different emissions scenarios was that some of those can be unrealistic, thus we believe it is more important to study more than one scenario. In addition, the SSP5-8.5 proves valuable in detecting the anthropogenic radiative forcing signal. We have added this in the discussion of our results:

- Lines 353–358: "In particular, we deemed it important to go beyond the worst-case scenario (SSP5-8.5), and also look at the implications of lower warming levels. **While most of the results presented indicate a stronger signal for the highest emission scenario (SSP5-8.5), this scenario has been deemed unrealistic. Therefore, we emphasize that our results should be interpreted with consideration of various scenarios.**"

L283. SSP5-8.5 scenario

We have corrected the typo. Thanks.

Table A2-A3. It could be helpful to add the periods (the year ranges) used for historial and SSP scenarios also here.

Thank you. We have added the periods.

**References**

Rudeva and Gulev (2007):

https://journals.ametsoc.org/view/journals/mwre/135/7/mwr3420.1.xml

Additional References:

Eiras-Barca et al. (2018): https://esd.copernicus.org/articles/9/91/2018/

Dacre et al. (2019): https://journals.ametsoc.org/view/journals/hydr/20/6/jhm-d-18-0175_1.xml

Zhang et al. (2018): https://agupubs.onlinelibrary.wiley.com/doi/full/10.1029/2018GL079071

Lim and Simmonds (2002): https://journals.ametsoc.org/view/journals/mwre/130/9/1520-0493_2002_130_2188_ecdits_2.0.co_2.xml

Ullrich et al. (2021): https://gmd.copernicus.org/articles/14/5023/2021/

McClennyet al. (2021): https://agupubs.onlinelibrary.wiley.com/doi/10.1029/2020JD033421

Collow et al. (2022): https://agupubs.onlinelibrary.wiley.com/doi/full/10.1029/2021JD036155

**2. Response to comment by Anonymous Referee #2**

This manuscript aims to evaluate the current climatology and assess changes under future climate scenarios of the concurrence between atmospheric rivers and extratropical cyclones undergoing explosive development – frequently referenced as explosive cyclones – in the North Atlantic. Being the explosive development of extratropical cyclones and atmospheric rivers crucial in driving extreme weather in the mid-latitudes, this topic is relevant, deserves to be investigated and it fits the scope of the Earth System Dynamics journal.

The manuscript is well-structured and well written and it is pleasant to read. It applies well-known datasets and detection and tracking methods previously published and discussed in the literature. However, in my opinion, some points are too succinct and need further details and explanations before the manuscript is accepted for publication.

First of all, we would like to thank the anonymous reviewer for the helpful comments on the manuscript.

Some points that need further clarification are:

- The authors use TempestExtremes Code for Detecting and Tracking Extratropical Cyclones and Atmospheric Rivers (ARs) for the North Atlantic region [25-65ºN; 80◦W-10◦E]. In Appendix "A4 Number of Cyclones and ARs detected in ERA5 and CMIP6" the words Cyclone and track are used as synonyms; in Figure 1 "EC track density climatology" is presented and "Units are the number of cyclones per 1.5◦ spherical cap per month"; and in

Figure 2 "AR frequency climatology" is presented with "Units are the percentage of time-steps detected as AR". Before a climatology can be presented and discussed, a clarification must be presented, and methods should detail the definitions and how this has been computed. In both cases, the authors must clarify if the systems are being tracked and considered or if the systems' timesteps are considered independently. All the processes to produce the cyclones' tracks and ARs datasets must be better explained. From my understanding, the systems are considered, but this is not clear from the discussion and captions of Figures 1, 2 and 8.

This is correct (the systems are being tracked and considered), we agreed the methodology needs clarification. We will ensure that the text clearly states what is being referred to in each case and will harmonise the nomenclature for consistency. Additionally, we have expanded the Methods section to describe how the climatologies presented in Figures 1 and 2 are calculated. In the main body of the manuscript (including Figures 1 and 2), we refer to the time steps of the tracked systems (the systems' time steps are considered independently). Only in Appendix A4 we do refer to the total number of tracked systems (the systems are being tracked and considered).

We have clearly specified whether we refer to **'cyclone tracks/AR tracks'** or **'cyclone timesteps/AR timesteps'** in all instances.

- **Figure 1 shows the number of cyclone timesteps detected within a 3deg spherical cap per month, hereafter referred to as cyclone track density, for ECs and non-ECs in both ERA5 and CMIP6 models.** These results agree with Priestley et al. 2021 for CMIP6 and Zappa et al. 2013 for CMIP5 track densities despite using different tracking algorithms. Our tracking method shows some differences between ERA5 and CMIP6 models (Figure 1c,f). **CMIP6 models underestimate the cyclone track density along the North Atlantic storm track, particularly east of Newfoundland, south of Greenland, and in the North Sea. For non-ECs, CMIP6 show a northward shift of the storm track with lower cyclone track density in the south and higher density in the north of the domain.**
- Figure 2a shows the **percentage of timesteps with a detected AR for ERA5, hereafter referred to as AR frequency**. Our tracking methodology accurately reproduces the AR frequency when compared with Guan et al. 2015. The AR frequency in the historical simulations of CMIP6 models is higher compared to ERA5 and exhibits a southward shift, with more AR timesteps detected primarily in the lower midlatitudes (Figure 2b,c).
- Title Appendix A4: "**Number of cyclone tracks and ARs tracks detected in ERA5 and CMIP6".** Caption**: "Number of EC and non-EC tracks detected in each dataset.**"

- The North Atlantic region [25-65ºN; 80◦W-10◦E] is considered. The authors should discuss the artefact over the western boundaries of the domain. A buffer area should be considered for the identification and tracking of the systems.

Thank you for highlighting this issue. We acknowledge the artefact in the western boundary of the domain in cyclone climatology. We identified that the cyclone tracking algorithm creates this issue specifically at the western boundary, where cyclones move eastward as they enter the domain. It creates stationary "artefact" cyclones that have their MDP along the boundary. To address this and ensure it does not affect our results, we applied a 10º buffer zone at all boundaries of the domain. The new tracking domain is 15-75ºN, 90ºW-20ºE, while for the

analysis, only the time steps of tracks within the original domain [25-65ºN, 80ºW-10ºE] are considered.

We found that this issue was impacting the concurrence results in Figures 4 and 5, where the peak of concurrence was initially 6 hours after the MDP. By correcting this issue and adding the buffer zone, the peak of concurrence now aligns with the MDP. This correction brings our results in line with those of Eiras-Barca et al. (2018). The previous shift in the peak of concurrence was due to the "artefact" cyclones having the MDP at the boundary meaning that these cyclones had the MDP at the first time steps of their tracks. This shifted the curves in Figures 4 and 5 to the right as the "artefact" cyclones were adding a bias only to the times after the MDP. All results that depend on the tracking have been updated.

- The method to identify the concurrences of Extratropical Cyclones and ARs also needs further explanation: it is presented through Figure 3 and the Xynthia case study, but this example elucidates the doubts and need for clarification on the methods. From this example, five timesteps are consistent with the concurrence of the cyclone under explosive development and the occurrence of the AR. The shaded areas in Figure 3 that depict the regions identified as ARs should have some correspondence with the cyclone track and should be described in the text as well. It is not clear to me how many times cyclone Xynthia and the concurrent AR are considered for the climatological assessment. I would say Figures 1 and 2 correspond to timesteps – and not Cyclones/AR.

We have modified the description of the methodology to make it as clear as possible. We have modified Figure 3 (see also reply to Mika Rantanen) to make it more useful to understand the methodology. As you suggested, we have included in Figure 3 which AR (shades) correspond to which time step in the cyclone track (crosses). Each cyclone time step and the concurrent AR (or not) is only used once in the climatology assessment or further in the following results sections (same for Xynthia).

[Figure]

- As mentioned previously, the method to identify the concurrences (Ln 130-134) must be further detailed. Please discuss the choice of the Maximum Deepening Point (MDP). An explanation should be given for the choice of the 1500 km threshold. It is not clear if a sensitivity analysis was performed, nor if this metric is constant for all cyclone's sequential timesteps. To the best of my understanding, each detected AR candidate may have more than one grid point being detected as AR in the same timestep. Certainly, the authors have considered this and all these aspects should be presented and discussed in the methods section. Additionally, how this method differs from Eiras-Barca et al. (2018) should be highlighted.

Thank you once again for your insightful comment. We have revised the text in the Methods section for clarity.

We selected the maximum deepening point (MDP) as a time reference because it allows us to better assess the influence of ARs on cyclone cyclogenesis. Additionally, using the MDP makes our results directly comparable to those of Eiras-Barca et al. (2018). We have specified this in lines 145–147.

Regarding the choice of the 1500 km threshold, I kindly refer you to our response to the first comment in our reply to Mika Rantanen. As you noted, our methodology aims to align as closely as possible with that of Eiras-Barca et al. (2018) to ensure comparability. The main difference lies in the tracking methodologies for ARs and cyclones, as we use different tracking algorithms but still, those algorithms share similar configuration parameters. However, in terms of cyclone/AR concurrence detection, explosive versus non-explosive cyclone classification, and the calculation of the MDP as a time frame, we follow the same methodology. We have emphasized all these points in the Methods section:

- Lines 149–156: "Subsequently, we determine whether a specific timestep of an extratropical cyclone (EC or non-EC) is linked to an AR by detecting at least one grid point classified as an AR within 1500 km from the centre of the cyclone. **Hence, each detected cyclone may have more than one grid point detected as an AR. This 1500 km radius is consistently applied across all time steps of the cyclone tracks. By selecting a 1500 km radius, our methods align with those of (Eiras-Barcas et al. 2018), with the primary difference between the two methods being the AR and cyclone tracking algorithms used. We consider that moisture brought by an AR may influence the cyclone within this radius by delivering moisture to the warm conveyor belt (WCB) or feeder airstream (Dacre et al. 2019). Most of the identified ARs are located in the southeastern quadrant of the cyclone (Supplementary Figures S1 and S2), which maximizes the probability that the AR and cyclone are dynamically linked through these two components.**"

- The clarification of these methodological aspects is vital for the discussion of the results: authors should clearly state if this study evaluates "the concurrence of ECs and ARs in the ERA5 reanalyses and we compare it with those obtained in climate models" considering only once each EC and AR.

We believe that the revisions made in response to the reviewer's first point, along with the updates to the methods section, now adequately address this concern.

- All the analysis is performed for the extended winter period (October to March). This should be indicated in the figure captions. Please clarify, in the methodology, how the inter-seasonal

variability is defined if only the extended winter season is considered. I suppose the authors mean interannual variability.

Thanks again for helping us to explain better the methodology. We have added in Figure captions that the period analyzed is extended winter (ONDJFM).

We mean interannual variability (variations between extended winters in different years), we have changed interseasonal for inter-annual everywhere in the manuscript. The interannual variability is defined as the standard deviation among the 30 years of each period and we have added this in the methodology section:

 - Lines 164–165: "**Additionally, we assess the inter-annual variability of the concurrences by calculating the standard deviation at each time step of the rate of coincidence between ECs and ARs over the 30 extended winter seasons (ONDJFM).**"

- CMIP6 models' information and discussion of results: additional detail should be included for the choice of one single member for each model and not the ensemble – how this particular member has been selected and how this choice may affect the final results. This should be included in the methodology and the discussion. Please refer to whether one may state that model X overestimates/underestimates the results or if model Y is more adequate for the analysis if only one member has been used. Please also discuss if the biases quantification is reliable. To the best of my knowledge, this assessment is not enough to make a comparison between models. A multimodel ensemble framework with varied combinations of GCMs is extremely useful and allows for reducing the uncertainty in climate projections for future scenarios and for a tendency assessment, but it can hardly be used to intercompare models when only one member is used. Please, define "the internal variability of the datasets" (ln 161) in the methodology section and how it is assessed in this manuscript.

The choice on the number of CMIP6 models and ensemble members is explained in lines **86– 91,** and the discussion of its limitations in the conclusions section lines **347–357**. We were restricted to using one member per model because the other ensemble members did not have the necessary variables to calculate IVT, which is essential for studying ARs. In essence, we used all available members from CMIP6 models that had the required variables for the historical period and the three scenarios.

We acknowledge, both here and in the manuscript, that the limited number of members is a limitation. For this reason, we assess changes between the present and future using the multi-model mean of the ensemble (ensemble of 6 members from 6 different models). We also agree that stating a particular model overestimates or underestimates results may not be appropriate when using only one member per model. We have revised this wording in the text and avoided using words such as overestimate and underestimate. We have added:

- Methods Section Lines 89–91: "**This limitation in the analyzed data prevents a complete assessment of model performance, as only a single member from each model is used, and a multi-member ensemble would be necessary for a more robust evaluation of model uncertainty. For this reason, we evaluate the results using the multi-model mean of the ensemble.**"
- Results section Lines 191–193**: "We emphasize that caution is needed when interpreting these model performances, as they may be influenced by internal variability due to the use of only a single member."**

- Conclusions section lines 350–352: "**We acknowledge that using only one member per model does not facilitate a comprehensive model intercomparison; more members for each model would be needed to adequately assess model uncertainty, specifically the biases of the models relative to reanalysis data.**"

In line 161, we refer to "the internal variability of the datasets" as the model's spread, or in other words, the spread within the multi-model ensemble. We have clarified this in the manuscript:

- Methods Section Lines 163–165: "**We further evaluate the internal variability of the CMIP6 concurrences by analyzing the spread within the multi-model ensemble. Additionally, we assess the inter-annual variability of the concurrences by calculating the standard deviation of the rate of coincidence between ECs and ARs over the 30 extended winter seasons (ONDJFM).**"
- Results Section Line 182–183: "CMIP6 models reproduce similar concurrence variability as ERA5 and differences in concurrence rate between them and ERA5 are within the internal variability, **computed as the spread of the multi-model ensemble.**"

- It would be useful if the results presented in the Appendix should be accompanied by a short description and discussion. Please avoid using expressions like "little change" (ln 190-191) or "lower-than-expected increase" (ln 194)– please quantify. A percentage could be added to tables.

Thanks for the comment, we have included the percentual change of the tracks in the Appendix Tables A2 and A3 and added a description and improved the information in the Appendix:

"Section 3.1 describes how cyclones are detected, tracked and finally classified as ECs or non-ECs. The result of this process is the track of each cyclone, in the following table we summarize the number of individual ECs and non-ECs tracks for each dataset and the percentual difference from the historical period:"

"Section 3.2 describes how ARs are detected, tracked. The result of this process is the track of each AR, in the following table we summarize the number of individual AR tracks for each dataset and the percentual difference from the historical period:"

We also have reviewed the text in (ln 190-191) and (ln 194) to avoid these expressions.

- Ln 200 – this sentence deserves additional information or a reference. It is out of context in this paragraph. These would be relevant results but evidence must be shown.

We separated this into another paragraph and provided additional context:

- Lines 227–235: "**The increase in the rate of coincidence between cyclones and ARs is partly the result of a decrease in the total number of cyclone time-steps and a direct increase in the number of concurrent cyclone/AR time-steps. To clarify this, we calculated the absolute number (including all CMIP6 models) of cyclones concurrent with ARs at the MDP and it increases across all scenarios: 4.6%, 7.1% and 6.3% for SSP1-2.6, SSP2-4.5 and SSP5-8.5 respectively. In addition, the absolute number of cyclone time-steps also at the MDP decreases: -3.6%, -5.2% and -12.3% for SSP1-2.6, SSP2-4.5 and SSP5-8.5 respectively. Hence, the combined effect of an increase in AR occurrence and a decrease in the number of cyclones, the last one especially relevant for SSP5-8.5, explains the rise in the rate**

**of coincidence as the level of warming increases (Figure 5a,b). These findings suggest that changes in the characteristics of ARs, cyclones, or their interactions may be driving the observed changes, rather than the result being merely a statistical artefact of more cyclones and ARs occurring individually.**"

- Conclusions: please discuss what is the novelty, for the present period, from the literature. The sentence "The fact that AR concurrences are larger after the MDP suggests that mature ECs (when they are deeper) can facilitate the formation of ARs in their surroundings" (ln 272-273) deserves to be further discussed and justified. Firstly, it is well known that the detecting and tracking methods still have large uncertainty in detecting the absolute minimum central pressure of an extratropical cyclone; secondly, the difference should be quantified; finally, and most importantly, the only conclusion that these results allow us to obtain, in this state, is that additional AR are detected – we cannot state that they only formed at that particular timestep.

Thank you for your comments. We have emphasized the key differences and novelty of our results. In lines (ln 272-273), we intended to convey that as more ARs are concurrent with ECs around the MDP, this suggests that ARs are more likely to occur when the EC is at its maximum deepening stage. This finding is supported by other studies, such as Zhang et al. (2018) and Eiras-Barca et al. (2018). This result holds true for both the historical period and all future scenarios. We have made the following changes:

- Lines 307–311: "For the present period, in ERA5 nearly 65\% of the Explosive Cyclones (ECs) are associated with an AR at the Maximum Deepeing Point (MDP). This higher rate of coincidence around the MDP indicates that ARs are more likely to occur when the EC is at its peak deepening stage. Despite some biases in the magnitude of the coincidence rate, CMIP6 models exhibit qualitatively similar concurrence rates to ERA5. Conversely, the evolution of the concurrence rate over the cyclone's lifespan is less pronounced for non-Explosive Cyclones (non-ECs)."
- Lines 331–335: "Our results for concurrence rates of cyclones and ARs in the present climate are broadly consistent with Eiras-Barcas et al. 2018. One key difference is that they calculated the MDP using a 24-hour time window, while we used a 6-hour window. These differences in time intervals, as well as detection and tracking configurations for cyclones and ARs, may account for the minor differences in concurrence rates. Furthermore, our study uses the latest ECMWF reanalysis, ERA5 (Hersbach et al., 2020), whereas their study used the earlier ERA-Interim dataset."

We agree and acknowledge that tracking algorithms come with inherent uncertainties. We have quantified the number of individual cyclone tracks detected (Table A2 in Appendix A4) and have highlighted and expanded these results in the text. As mentioned in a previous comment, we will also include the percentage changes in the table. While our results have not identified clear changes in cyclone track counts, we believe that despite uncertainties introduced by tracking algorithms, our results (among other findings) confidently indicate an increase in the coincidence between cyclones and ARs, as well as an increase in ARs and their intensity.

References:

Eiras-Barca et al. (2018): https://esd.copernicus.org/articles/9/91/2018/

Zhang et al. (2018): https://agupubs.onlinelibrary.wiley.com/doi/full/10.1029/2018GL079071

**3. Response to comment by Anonymous Referee #3**

**Summary**

In this study, the authors investigated the future changes of compound explosive cyclones (ECs) and atmospheric rivers (ARs) in the North Atlantic using the simulation data from six CMIP6 models. Different from previous papers, this study focused on the future changes of compound ECs and ARs, which usually develop rapidly due to the strong diabatic feedback and are closely related to extreme precipitation and wind. The authors found that there is a significant and systematic future increase in the EC-AR concurrences, especially over Western Europe in the high-emission scenario. Overall, this study investigated the future changes of ECs and ARs from a novel perspective and the paper is well written and organized. However, I have some major concerns, especially for the methods, and general comments listed below for the authors' consideration.

First of all, we would like to thank the anonymous reviewer for the helpful comments on the manuscript.

**Major Concerns:**

(1) My biggest concern is the sensitivity of the conclusions in this study to the AR and cyclone tracking methods. For example, there are many different AR detection methods with large differences as summarized in some papers from the Atmospheric River Tracking Method Intercomparison Project (ARTMIP; e.g., O'Brien et al. 2020, 2022; Shields et al. 2019). If a different AR detection method or a different cyclone tracking method is used, will that have a significant impact on the conclusions about the EC-AR concurrences?

Thanks for this comment. We are aware and acknowledge that tracking algorithms can show significant differences (see the last paragraph in the conclusions). Among other reasons detailed in the manuscript, we choose to use the TempestExtremes algorithm to detect and track both cyclones and ARs because the cyclone and AR climatologies shown in Figures 1 and 2 sit in the middle of the results for the climatologies of cyclones and ARs shown in the tracking comparison studies of Neu et al. (2013) (for cyclones) Collow et al. (2022) (for ARs). Despite that, we tested the sensitivity of our results by comparing those with Eiras-Barca et al. (2018). They use two different tracking algorithms and ERA-Interim, and our results for ERA5 of the rate of concurrence (Figure 4 a,b) sit in between their results for the two algorithms (Eiras-Barca et al. (2018), Figure 3a). We are aware that the absolute values of concurrence rates can be sensitive as shown when compared to Eiras-Barca et al. (2018). Still, the changes between historical and future scenarios need to be tested with the same methodology. The main objective of our paper is to asses future changes in the rate of concurrence of cyclones and ARs and for that same methodology should be applied to both present and future periods. Given this focus, we believe that differences among periods are less sensitive to the tracking algorithms. This is supported by Zhang et al. (2024) in Figure 7 where they show larger agreement in the AR frequency future changes than in the climatologies among different AR tracking methods. A similar result is shown by O'Brien et al. 2022 (Figure 1), where differences in AR climatologies among tracking algorithms are large but those agree in future AR frequency trends.

(2) Section 3.3: "an extratropical cyclone (EC or non-EC) is linked to an AR by detecting the presence of an AR within a 1500 km of the cyclone center" (lines 133-134). It may be oversimplified to use 1500 km distance to determine the concurrence between cyclone and AR. If a cyclone and an AR are dynamically associated with each other, the AR is usually located over the south to southeast side (the position of the low-level jet stream ahead of the cold front) of the cyclone center. In that case, it makes sense to define the concurrence if an AR exists within 1500 km. However, if an AR is located over the north or northwest side of a cyclone center, I don't think it is reasonable to say that the AR is dynamically associated with the cyclone even if the distance is within 1500 km.

Thank you once again for your insightful comment. Regarding the choice of the 1500 km threshold, I kindly refer you to our response to the first comment in our reply to Mika Rantanen:

"The main reason to use the threshold of 1500 km for the radius detection of ARs in the surroundings of a cyclone is to be able to compare our results with the study of Eiras-Barca et al. (2018), where they evaluated the concurrence of ARs and cyclones with this radius for the historical period but using different tracking algorithms. In addition, we believe that the moisture brought by an AR (even if this is located further than 900 km) still influences the cyclone as in many cases is not the AR delivering its moisture directly to the cyclone centre but is the WCB or the feeder airstream that connects the enhanced moisture area of the AR with the cyclone and ultimately enhancing its intensification (Dacre et al. 2019). For this reason, ARs within 1500 km of the cyclone can contribute to its deepening as other airflows within the cyclone transport the moisture that potentially contributes to intensification."

We have analyzed the distribution of the closest points of atmospheric rivers (AR) around the cyclone center (Fig. R1) and found that most of them fall within the east-southeast quadrant of the 1500 km circle. This is expected, as most ARs that are dynamically associated with a cyclone are located to the southeast of the cyclone center, linked to the WCB and feeder airstream. This pattern is further illustrated in Figure A2, where the histogram shows that most ARs are in the southeast quadrant. If we expand this radius to 2500 km, there is a shift in AR occurrences toward the southwest and northwest quadrants of the cyclone (Fig. R2), making it less likely that these ARs are dynamically linked to the cyclone. Therefore, we conclude that 1500 km represents a good compromise between maximizing the number of possible cases (statistics) and selecting those that are dynamically linked (dynamics).

We have included these two figures in the Supplementary material (Fig. S1 and S2), as well as included the following justification in the article:

- Lines 149–156: "Subsequently, we determine whether a specific timestep of an extratropical cyclone (EC or non-EC) is linked to an AR by detecting at least one grid point classified as an AR within 1500 km from the centre of the cyclone. **Hence, each detected cyclone may have more than one grid point detected as an AR. This 1500 km radius is consistently applied across all time steps of the cyclone tracks. By selecting a 1500 km radius, our methods align with those of (Eiras-Barcas et al. 2018), with the primary difference between the two methods being the AR and cyclone tracking algorithms used. We consider that moisture brought by an AR may influence the cyclone within this radius by delivering moisture to the warm conveyor belt (WCB) or feeder airstream (Dacre et al. 2019). Most of the identified ARs are located in the southeastern quadrant of the cyclone (Supplementary**

**Figures S1 and S2), which maximizes the probability that the AR and cyclone are dynamically linked through these two components.**"

In addition to the sensitivity tests already shown here, we underscore that there are examples where the closest AR grid point to a cyclone can sometimes be located to the north or northeast of the cyclone and still have a dynamical association with it. The shape of the AR, particularly its front part, often takes on a sickle shape that curls around the cyclone. As a result, it may not be uncommon for the closest AR grid point to be in these quadrants and have a dynamical association with the cyclone. Next, we provide an example of the AR "sickle shape" (from ERA5, 14th January 2009 at 18 UTC):

[Figure]

Figure R1. Example of the AR sickle shape from ERA5, 14th January 2009 at 18 UTC).

(3) Fig.4, the description in Section 4, and many other places throughout the manuscript: It is a smart way to use the maximum deepening point (MDP) as a reference point. However, the lifetime of extratropical cyclones has a large variability, varying from a couple of days to over one week. So it is very arbitrary to say that 36 hours before MDP (-36 h) is "the initial stages of cyclone formation", +36 h is "the dissipation stages of the cyclones", and from -36 h to +36 h is "the lifetime of the ECs".

Thanks for this comment, other reviewers also pointed this out. We agree that formation/dissipation stages might not be appropriate as we analyse the cyclones +/-36 hours from the MDP. We have changed this and modified the text accordingly to avoid referring to formation/dissipation stages in this context.

(4) The domain for analysis is 25N-65N and 80W-10E in this study (line 74). Did the authors use only the data within this domain for AR/cyclone tracking and EC-AR concurrence determination? If yes, there will be a boundary issue, especially for the cyclones and ARs near the boundaries, since both AR and cyclone detections have thresholds for moving distance and existing time. For example, in Fig.1 the EC track density is unreasonably high along the western boundary of the domain; in Fig.8 the rate of coincidence tends to be very small close to the boundaries. My concern is that the boundary issue may have impacts on the conclusions, especially for Western Europe, the area around the south tip of Greenland, and other places close to the boundary.

Thank you for this important comment. This point has also been raised by another reviewer; please refer to our response to the second comment from Reviewer #2:

"We acknowledge the artefact in the western boundary of the domain in cyclone climatology. We identified that the cyclone tracking algorithm creates this issue specifically at the western boundary, where cyclones move eastward as they enter the domain. It creates stationary "artefact" cyclones that have their MDP along the boundary. To address this and ensure it does not affect our results, we applied a 10º buffer zone at all boundaries of the domain. The new tracking domain is 15-75ºN, 90ºW-20ºE, while for the analysis, only the time steps of tracks within the original domain [25-65ºN, 80ºW-10ºE] are considered.

We found that this issue was impacting the concurrence results in Figures 4 and 5, where the peak of concurrence was initially 6 hours after the MDP. By correcting this issue and adding the buffer zone, the peak of concurrence now aligns with the MDP. This correction brings our results in line with those of Eiras-Barca et al. (2018). The previous shift in the peak of concurrence was due to the "artefact" cyclones having the MDP at the boundary meaning that these cyclones had the MDP at the first time steps of their tracks. This shifted the curves in Figures 4 and 5 to the right as the "artefact" cyclones were adding a bias only to the times after the MDP. All results that depend on the tracking have been updated."

(5) For the horizontal resolution, ERA5 is 0.25 degrees while the six GCMs are quite different, varying from ~0.7 degrees to ~2.0 degrees. Did the authors interpolate the data to a common grid before analysis? Will the different horizontal resolutions have any impact on the conclusion? For example, the AR intensity is defined as the maximum IVT, but the different horizontal resolutions may have an impact on the maximum IVT across different models and ERA5. As a result, some differences across different models and ERA5 might be a data resolution issue, not the real model bias.

We did not interpolate the data to a common grid; all analyses were conducted at the native resolution of the datasets. This approach is consistent with how Ramos et al. (2016) handled different grid resolutions in CMIP5 when analysing ARs or similar to O'Brien et al. (2022) and Zhang et al. (2024) that apply the AR tracking algorithms to the CMIP models native grids. We acknowledge that differences in model resolution introduce uncertainty, but we consider this an inherent uncertainty of the models themselves. Moreover, when analysing differences between periods, we compare the multi-model means, which is equivalent to comparing each model to itself and then averaging the differences. We never perform a direct comparison across different models. Our main goal here is to assess changes between historical and future scenarios of AR intensity and not model evaluation with ERA5.

We acknowledge that ERA5 shows the largest differences with the historical models, so we performed a sensitivity test where the SLP and IVT fields for ERA5 are interpolated to a 1º, 1.5º and 2º regular lon-lat grid (similar range to the CMIP6 models resolutions). Then the same methodology described in the manuscript is applied to the regrided ERA5. We analysed the sensitivity of the resolution in ERA5 to the concurrence of ARs with ECs and non-ECs as it is done in Fig. 4 a,b (Fig. R2, and Fig. R3 of the Supplementary material). The rate of coincidence decreases as we reduce the resolution of ERA5 for both EC and non-ECs in the same way. The spread of the regridded ERA5 (1º,1.5º and 2º) is slightly smaller than the spread of the CMIP6 models, but the CMIP6 models do not show lower rate of coincidence for the coarser model

resolution. For example, EC-Earth, the model with the highest resolution, shows the lowest concurrence rate of CMIP6 models, or MPI-LH with the lowest resolution has higher concurrences than average. With this sensitivity test we can say that the CMIP6 model spread is not driven by the model resolution since they show different signals. The model spread is driven by the inherent nature of each model (including its resolution), as well as internal variability, and we study it accounting the resolution as a characteristic of each model.

[Figure]

Figure R2. Same as Figure 4a,b of the manuscript but with additional curves for ERA5 interpolated to different resolutions (1º,1.5º and 2º).

Finally, we recognize that maximum IVT is sensitive to resolution. However, we chose this variable because it is not dependent on the AR shape. An alternative would be to use the mean IVT within the AR, but AR shape (particularly the size) is also sensitive to model resolution. As a sanity check, we also performed a resolution sensitivity test to the IVT-max results by reproducing Fig. 6 a,b with the regrided ERA5 (Supplement Figure R3, and Fig. S4 of the Supplementary material). The IVT-max values decrease when lowering the resolution in ERA5 for both ECs and non-ECs, the CMIP6 models more or less align with this since higher resolution models show higher IVT-max than the CMPI6 mean and the models with lower resolution show lower IVT-max than the CMPI6 mean. Despite of that, the IVT-max CMIP6 model spread is at least 3 times larger than the regrided ERA5 (1º,1.5º and 2º). Again the model spread is driven by the inherent nature of each model (including its resolution) rather than only the resolution of the model.

This sensitivity analysis are included in the Supplement of the manuscript and mentioned in the relevant results sections.

[Figure]

Figure R3 Same as Figure 6a,b of the manuscript but with additional curves for ERA5 interpolated to different resolutions (1º,1.5º and 2º).

Finally, we would like to emphasize that we do not perform a direct comparison across different models. Our main goal here is to assess changes between historical and future scenarios models mean and not model evaluation with ERA5 and we consider resolution as an inherent property of the models themselves.

(6) A few concerns/questions about the AR and cyclone tracking methods.

Line 118: In addition to detecting ridges in the IVT field, there is an IVT minimum threshold of 250 kg/m/s. However, the IVT values have large variability from low to high latitudes. Will the 250 kg/m/s minimum threshold be too high for the ARs at high latitudes, like the area around the south tip of Greenland, near or higher than 60N?

The 250 kg/m/s minimum threshold was added as a sanity check as it is one of the most used thresholds, but the detection threshold used is the laplacian of the IVT, less sensitive to latitude differences. Anyway, 250kg/m/s is not a high threshold for ARs at least under 65N (our case), see ARs IVT climatologies in Thandlam et al. (2022).

Line 118: The AR candidates should have an area larger than $4 \times 10^5$ km$^{-2}$. Are there any requirements for the AR shape? ARs are usually defined as a long and narrow corridor of strong water vapor transport.

In this case, the geometrical requirement is the area threshold (also a very common threshold in AR detection algorithms). The TempestExtremes algorithm does not have a wide/length requirement, even the many of the AR tracking algorithms have a length requirement this is not the case for all (Shields et al. 2018).

Line 119: "... concatenated if at least one grid point …" Similar to major comment (5), if the data were not interpolated to a common grid, it is a concern since models have quite different horizontal resolution, which means the "one grid" threshold is different across different models (~0.7 degrees to ~2.0 degrees) and ERA5 (0.25 degrees).

We acknowledge that this can be a source of uncertainty, but in our answer to comment (5) we consider the model resolution as an inherited source of uncertainty of the model itself and show a resolution sensitivity analysis.

In addition to this sensitivity analysis, we show in Table S1 of the Supplementay material the number of AR tracks and EC and non-EC tracks for the regrided ERA5 following the same methodology as in Table A2 and A3. The number of ARs tracks only shows a 5% difference when regridding ERA5 to 2º, suggesting that the "one grid" threshold for the AR tracking has almost no effect on the results. The total number of EC and non-EC tracks show a larger decrease for ERA5 2º but not as much as for ERA5 1º and 1.5º, similar to what we observe between MPI_HR and MPI_LR. For the other models the differences in number of cyclone tracks is not proportional to the resolution.

| #Tracks | ERA5 | ERA5 1º | ERA5 1.5º | ERA5 2º | MPI_HR | MPI_LR | EC_Earth3 | NorESM | MIROC6 | CMCC_ESM2 |
|---------|------|---------|-----------|---------|--------|--------|-----------|--------|--------|-----------|
| AR | 1224 | 1225 | 1214 | 1165 | 1219 | 1286 | 1187 | 1235 | 1152 | 1186 |
| non-EC | 3200 | 3391 | 3240 | 3092 | 3046 | 2387 | 2927 | 3594 | 3424 | 3530 |
| EC | 1372 | 1307 | 1262 | 1158 | 1168 | 870 | 1283 | 1193 | 879 | 1076 |

Lines 91 and 92: "not exceed 6 GCD degrees" and "at least 12 GCD degrees". Are there any specific reasons for using 6 GCD and 12 GCD?

For these parameters, we used the default tracking setting from Ullrich et al. (2021). We now mention it in the Methods section: "**To identify extratropical cyclones, we recognize candidate "nodes" corresponding to local minima in the SLP field with the same set-up as in Ullrich et al. (2021).**"

**Minor Comments**

(1) Line 11: "worst-case scenario", I think it would be better to use high-emission scenario.

Thank you for your comment, we have changed worst-case scenario for high-emission scenario.

(2) Why did the authors select those six CMIP6 models while there are many other GCMs in CMIP6?

The choice on the number of CMIP6 models and ensemble members is explained in lines **86–91,** and the discussion of its limitations in the conclusions section lines **347–357**. We were restricted to using one member per model because the other ensemble members did not have the necessary variables to calculate IVT, which is essential for studying ARs. In essence, we used all available members from CMIP6 models that had the required variables for the historical period and the three scenarios.

We acknowledge, both here and in the manuscript, that the limited number of members is a limitation. For this reason, we assess changes between the present and future using the multi-model mean of the ensemble (ensemble of 6 members from 6 different models). We also agree

that stating a particular model overestimates or underestimates results may not be appropriate when using only one member per model. We have revised this wording in the text and avoided using words such as overestimate and underestimate. We have added:

- Methods Section Lines 89–91: "**This limitation in the analyzed data prevents a complete assessment of model performance, as only a single member from each model is used, and a multi-member ensemble would be necessary for a more robust evaluation of model uncertainty. For this reason, we evaluate the results using the multi-model mean of the ensemble.**"
- Results section Lines 191–193: "**We emphasize that caution is needed when interpreting these model performances, as they may be influenced by internal variability due to the use of only a single member.**"
- Conclusions section lines 350–352: "**We acknowledge that using only one member per model does not facilitate a comprehensive model intercomparison; more members for each model would be needed to adequately assess model uncertainty, specifically the biases of the models relative to reanalysis data.**"

(3) Line 101: "These results agree with Priestley and Catto (2022) and Zappa et al. (2013) …" It's worth noting that Zappa et al. 2013 used CMIP5, not CMIP6.

We have modified this in the text: "These results agree with Priestley and Catto (2022) **for CMIP6** and Zappa et al. (2013) **for CMIP5** track densities results despite using different tracking algorithms."

(4) Line 103: "Figure 1c,d", do you mean Figure 1 c and f?

Yes, we have corrected that.

(5) Line 134: delete "a" before "1500 km".

Thank you, we have correted the typo.

(6) Fig.1 and Fig.2: It would be very helpful to show the percentage difference in addition to the absolute difference of cyclone track density and AR frequency. Same for the other difference figures.

We tested changing the absolute difference for the percentage difference in these figures, but we believe that showing the absolute difference (CMIP6 - ERA5) makes this figures easier to interpret. The percentage difference (CMIP6 - ERA5 / ERA5) in some areas can show very large percentual difference which is harder to understand. This is because in these  areas the AR or cyclone climatology can be 0 or almost 0 and can make the percentual difference very large in this cases. That is why we think that it does not make sense to show it as percentage but better as absolute difference.

(7) Fig.2: The unit of AR frequency is % in the figure but the values are fraction (0.00~0.12).

You are right. We have fixed it, now it is in %.

(8) Fig.4: The difference across different lines (models) is not very clear. Maybe use different colors for different models?

We used a set of color-blinded fot the different lines in Figure 4 in order to make them more distinguishable.

(9) Line 144 and some other places: "MDP point". "Point" is redundant since MDP is maximum deepening point.

You are right, it is redundant. We have removed "point" after the MDP in all places.

(10) Line 149: "… favours the detection of an AR in its surroundings". "detection" is not suitable here, maybe change it to "existence".

We agree, we have rephrased this sentence, now we do not use "detection".

(11) Line 158: "the standard deviation of the rate of coincidence". Is that calculated using the coincidence rate at MDP or from -36 h to +36 h?

It is calculated using the coincidence rate at each time from -36 h to +36 h. Now it appears in the text like: **"Figure 4c shows the evolution of inter-annual variability as the standard deviation of the rate of coincidence between ECs and ARs over the 30 winter seasons at each time from -36 h to +36 h from the MDP."** In addition, this is also explained at the Methodology section.

(12) Line 173 and 175: 0.08 and 0.05 are the model biases of what? Coincidence rate? For the average of all models or the model with the maximum bias?

Yes, of the coincidence rate. It refers to the model with the maximum bias or as we refer in the text now as the "maximum model difference with respect ot ERA5". We have rewritn these sentences accordingly.

(13) Line 227: "The AR intensity for ERA5 is larger than any model for the historical period because ERA5's resolution is almost 4 times higher than the CIMP6 models, and attains larger values of IVT-max." So the difference of AR intensity between ERA5 and models might be a data resolution issue? This is the same question as my major concern (5).

Thanks for your comment, we agree and acknowledge that one of the main reasons for the difference in the IVT-max in this case (ERA5 vs CMIP6) might be driven by the resolution difference. We would kindly refer to our answer in the comment (5).

(14) Fig.6: Does the Non-EC AR means the ARs without an explosive cyclone or the ARs without any cyclone (no matter weak or explosive)?

It means the ARs associated with a non-explosive cyclone (non-EC), or in other words all the cyclones that are not categorized as Explosive Cyclones (EC) . The ARs analysed here are those associated either with ECs or non-EC.

(15) Fig.8e, there are many areas show a large increase in coincidence rate with model agreement. Why did the authors only emphasize the western Europe in the Abstract (line 12)?

We emphasize the Western Europe area in the abstract because it is potentially more relevant for the possible impacts in the society as being an area highly populated.

(16) Fig.8 a and b, the brown-green color map gives the readers the impression that the brown and green areas are opposite rather than low to high. I would suggest using a different color map.

Thanks for the suggestion. We have changed the color of maps in Fig.8 a and b, now instead of brown-green is only in scale of reds.

**References:**

Neu et al. (2013): https://journals.ametsoc.org/view/journals/bams/94/4/bams-d-11-00154.1.xml

Collow et al. (2022): https://agupubs.onlinelibrary.wiley.com/doi/full/10.1029/2021JD036155

Zhang et al. (2024): https://agupubs.onlinelibrary.wiley.com/doi/10.1029/2023JD039359

Thandlam et al. (2022): https://link.springer.com/article/10.1007/s00704-021-03776-w

Shields et al. (2018): https://gmd.copernicus.org/articles/11/2455/2018/

Ramos et al. (2016): https://agupubs.onlinelibrary.wiley.com/doi/full/10.1002/2016GL070634

Ullrich et al. (2021): https://gmd.copernicus.org/articles/14/5023/2021/

**4. Response to comment by Anonymous Referee #4**

The title of this manuscript does well to summarize what to expect. The overall quality of the manuscript is good. The writing is clear. The authors have carried out a substantial amount of application of existing Lagrangian tracking algorithms to reanalysis and climate model data, and then they have done some interesting sorting of the data. Ultimately the results suggest only a small signal amidst the noise of midlatitude storms. However, for the important issue of explosive cyclones, perhaps a null result is still useful. As ever, I do think we need to be cautious because there is always the lingering doubt about these model's ability to capture the physics of explosive cyclones.

I appreciate the author's choice on method of tracking ARs using the Laplacian of the IVT, so that they are not just picking up the thermodynamic signal. However, I have a fundamental issue with the way in which some concepts are explained in the introduction, and some questions about the interpretation of the results. These issues and questions are described below.

First of all, we would like to thank the anonymous reviewer for the helpful comments on the manuscript.

Major Comment:

Lines 37 – 41: This is a section in the introduction in which the authors seek to make a physical explanation for why the presence of atmospheric rivers (ARs) impact explosive cyclones (ECs).

However, I do not think these studies prove cause versus effect. I posit that in many, or perhaps half of the cases, it might be the case that rapidly intensifying cyclones have substantial upper-level forcing that drives more poleward transport of water vapor. This would lead to more ARs found in the surroundings of ECs, but the cause is not the upper-level circulation, not the latent heat release. (Isn't this substantiated by your result that more ARs are found to be associated with the cyclones after their maximum deepening point? – line 147.)

I want to make clear about my point: If the upper-level circulation is held fixed (e.g., in a modeling study for a single event or a baroclinic wave), then the storm intensity and intensification rate will increase with more water vapor (i.e., the presence of a stronger AR). However, that is different from saying that the presence of ARs leads to explosive cyclones. For me, the explanation provided by the authors in this section needs more nuance and explanation.

Relatedly, the papers being referenced in this section all state that their results "suggest" a relationship, but none of them claim it to be conclusive. So, I request that the authors add more caveats and details to this explanation. This would impact the introduction, the interpretation of results and the conclusions.

We thank the reviewer for their valuable feedback and agree with the concerns raised. In response, we have revised and extended lines 37–41 to reflect the suggested changes.

- Lines 35–45: The climatological relationship between ECs and ARs has been previously studied and the literature evidences that ARs are more often found in the surroundings of EC than non-ECs (Eiras-Barca et al., 2018; Zhang et al., 2019; Guo et al., 2020). **ARs are important sources of moisture for cyclonic systems, and it has been suggested that they can enhance cyclone deepening through moist diabatic processes (Zhu and Newell, 1994; Ferreira et al., 2016), such as cloud condensation (Pinto et al., 2009). In addition, ECs with ARs show larger moisture inflow and deepen more rapidly but do not show significant differences in low-level baroclinicity nor upper-level potential vorticity, suggesting that diabatic processes are important contributors to their intensification (Zhang and Ralph, 2021). However, the extent to which these moist diabatic processes, compared to other factors such as upper-level forcing, influence cyclone intensification can vary from case to case (Pfahl and Sprenger, 2016; Ginesta et al., 2024).**

Additionally, based on comments from other reviewers, we recognized an issue with the tracking of cyclones. Specifically, we did not account for a buffer zone, which affected our results regarding previous line 147 and the peak of intensity after the maximum deepening rate. After revising the plots, the peak perfectly alignes with the maximum deepening rate of the cyclones, consistent with previous studies and the theory outlined in the introduction.

Minor Comments:

Line 180: Figure 4 (and all similar plots): I suggest you replace h with the word hours to reduce any chances for confusion from a viewer.

We have changed *h* for *hours* in Figures 4, 5, 6, 7 and A1 in all x-axis labels.

Line 215: I am a bit puzzled by the AR intensity analysis in Section 5.2. In the methods section, you do a good job of explaining why the use of the Laplacian is important. Now you are back to working with IVT itself. Why? Given that storm forcing from latent heating (e.g., the change in diabatic potential vorticity) is related to the gradient of the heating, not the absolute value, this choice of defining AR intensity based on the absolute value should be explained in more detail.

The laplacian of IVT was used only for AR detection. We use the IVT itself because we want to quantify how much the intensity of the ARs will change in the different future scenarios. The IVT is the most used variable to study AR intensity, is well correlated with cyclone intensity and is also a proxy for the potential amount of precipitation (Ferreira et al. 2016; Guan et al. 2023). Our aim in this study is to assess changes in future scenarios of ARs and ECs, we acknowledge that our study has a limitation in giving a physical explanation for the intensification mechanisms between them, in this context studying the gradient of IVT would be a good way to do it. For our purposes, we believe that the IVT-max might be a better-fitting variable and will facilitate comparison with other studies of ARs in climate projections (Zhang et al. (2024)).

Line 247-8: Here you state:

"The results from ERA5 show the same behaviour for both types of cyclones but with lower intensity". Could you clarify this sentence to explain what intensity is referring to? Is it the intensity of the relationship or the intensity of the cyclones? If it is the intensity of the relationship, then perhaps you should also include a sentence or two here reminding the readers of the multiple reasons for potential biases in the models.

We agree that the original sentence was unclear. We have added:

- Lines 279–283: "**The results from ERA5 show similar behaviour for both types of cyclones when compared to the models (Fig. 7b,d). Before the MDP, the models tend to simulate lower SLP for ECs with ARs and higher SLP for ECs without ARs. After the MDP, the models generally simulate higher SLP for both ECs with and without ARs. For non-ECs, the models have higher SLP values after the MDP compared to ERA5. However, the ERA5 values fall within the ensemble spread of historical values, indicating that they are within the uncertainty range of the models.**"

Additional papers on water vapor and storm intensity that must be cited and discussed when discussing the results, given the nature of this manuscript:

Pfahl, S. and Sprenger, M.: On the relationship between extratropical cyclone precipitation and intensity, Geophys. Res. Lett., 43, 1752–1758, 2016 https://doi.org/10.1002/2016GL068018

Booth, J. F., Naud, C. M., and Jeyaratnam, J.: Extratropical Cyclone Precipitation Life Cycles: A Satellite-Based Analysis, Geophys. Res. Lett., 45, 8647–8654, 2018

https://doi.org/10.1029/2018GL078977

Sinclair, V. A. and Catto, J. L.: The relationship between extratropical cyclone intensity and precipitation in idealised current and future climate, Weather and Climate Dynamics, vol. 4, no. 3, pp. 567–589. doi:10.5194/wcd-4-567-2023, 2023

We thank the reviewer for the suggestions. We will cite them accordingly.

Additional References:

Guan et al. (2023): https://agupubs.onlinelibrary.wiley.com/doi/10.1029/2022JD037180

Ferreira et al. (2016): https://www.sciencedirect.com/science/article/pii/S1474706516000048

Zhang et al. (2024): https://agupubs.onlinelibrary.wiley.com/doi/10.1029/2023JD039359